# Intramuscular neutrophil-derived immunometabolic niches locally boost insulin-responsive GLUT4 translocation after muscle contraction

Weijian Chen and Makoto Kanzaki 

*Graduate School of Biomedical Engineering, Tohoku University, Sendai, Japan*

Handling Editors: Karyn Hamilton & Nima Gharahdaghi

The peer review history is available in the Supporting Information section of this article (https://doi.org/10.1113/JP290203#support-information-section).

**Abstract figure legend** Contraction-induced formation of a NET-based immunometabolic niche potentiates regional insulin sensitivity and GLUT4 translocation.

The schematic model illustrating how skeletal muscle contraction promotes a NET-based immunometabolic niche that enhances regional insulin sensitivity. Muscle contraction induces the release of myokines/exerkines (1), leading to neutrophil recruitment to the skeletal muscle microvasculature via CXCR2/CX3CR1 signalling (2). Recruited

**Weijian Chen** obtained his doctoral degree in biomedical engineering from Tohoku University. During his doctoral training he contributed to multiple publications examining skeletal muscle biology in response to exercise, with a particular focus on exercise-induced regulation of GLUT4 translocation, insulin-responsive signalling, post-exercise metabolic effects and immune components. His primary research interests centre on the beneficial effects of exercise on physiological and immunological function, including the regulation of myokines and muscle–immune crosstalk.

The Journal of Physiology

neutrophils form intravascular neutrophil extracellular traps (NETs) within the capillary network (3), which may alter the local perivascular microenvironment and insulin access. NET formation locally potentiates insulin signalling in adjacent myofibres, resulting in modest Akt (S473) and AS160/TBC1D4 (T642) phosphorylation and enhanced sarcolemmal GLUT4 accumulation. This spatially restricted immunometabolic niche provides a conceptual framework linking vascular immune responses to myofibre insulin action following muscle contraction.

**Abstract**  Exercise is well known to enhance insulin sensitivity in skeletal muscle, yet the underlying mechanisms remain incompletely understood. We have previously shown that neutrophil recruitment contributes to contraction-induced GLUT4 translocation and local myokine induction, but whether these immune cells also participate in the post-exercise increase in insulin sensitivity has been unclear. Here using GLUT4-EGFP transgenic mice and sciatic nerve-mediated *in situ* contraction of the hindlimb, with analyses focused on extensor digitorum longus (EDL) muscle, we demonstrate that neutrophil recruitment and subsequent formation of neutrophil extracellular traps (NETs) are crucial for the well-known post-exercise increase in insulin sensitivity. Two-photon imaging revealed that NET-like cell-free DNA (cfDNA) structures persisted for hours after contraction, forming spatially confined perivascular immunometabolic niches along the capillary meshwork. Strikingly insulin-stimulated GLUT4 translocation was preferentially enriched at these NET-rich sites, whereas DNase-mediated NET degradation eliminated cfDNA signals and abolished the contraction-induced enhancement of GLUT4 translocation, glucose uptake and attenuated AS160 (T642) phosphorylation under low-dose insulin. Our findings demonstrate that neutrophils are essential components of the mechanism underlying enhanced post-exercise insulin sensitivity involving, at least in part, the local formation of NETs. These NET-governed immunometabolic niches constitute a structural and spatial framework underlying the exercise-induced acute improvement of insulin-responsive metabolic efficiency in skeletal muscle.

(Received 28 September 2025; accepted after revision 3 February 2026; first published online 22 February 2026)
**Corresponding author**: M. Kanzaki: Graduate School of Biomedical Engineering, Tohoku University, 6-6-04-110, Aramaki, Aoba-ku, Sendai, 980-8579, Japan.    Email: makoto.kanzaki.b1@tohoku.ac.jp

## Key points

- Neutrophil extracellular traps (NETs) establish spatially confined immunometabolic niches that are indispensable for the post-exercise increase in insulin sensitivity.
- High-resolution imaging revealed that insulin-stimulated GLUT4 translocation is markedly enhanced predominantly in NET-rich perivascular regions, indicating a spatially restricted mechanism of post-exercise insulin sensitization.
- DNase-mediated degradation of NETs abolished this enhancement, establishing their essential role in local insulin-responsive GLUT4 translocation.
- These NETs are formed by neutrophils rapidly recruited to skeletal muscle after contraction and deposited along the capillary network.

## Introduction

The capacity of exercise to enhance insulin sensitivity in skeletal muscle has long been recognized (Cartee et al., 1989), yet the fundamental principles governing this effect have remained elusive despite decades of intensive investigation (Cartee, 2015; Richter et al., 2025). Studies at multiple biological levels have shown that this effect is multifactorial. On the molecular scale exercise induces a sustained phosphorylation of

the Rab-GTPase-activating proteins (GAPs), including AS160/TBC1D4 and TBC1D1, which preserve GLUT4 trafficking competence beyond the period of muscle contraction (Hatakeyama et al., 2019; Nedachi et al., 2008). Experiments on isolated single fibres have further demonstrated that prior contraction leaves muscle cells in a 'primed' state, enabling a stronger GLUT4 response to subsequent insulin stimulation (Hatakeyama & Kanzaki, 2017). *In vivo* studies in rodents and humans have consistently shown that a single bout of exercise enhances

skeletal muscle insulin sensitivity for 24–48 h (Funai et al., 2010; Oki et al., 2018a; Oki et al., 2018b; Treebak et al., 2009). Thus the post-exercise increase in insulin sensitivity constitutes a distinct adaptive state that arises after muscle activity, priming skeletal muscle for enhanced insulin-stimulated GLUT4 translocation and glucose uptake. This transient enhancement of insulin sensitivity is widely regarded as a central mechanism by which exercise ameliorates insulin resistance, and elucidating its underlying principles is essential for understanding the metabolic benefits of exercise (Zierath et al., 2024).

Recent studies have implicated immune cells, particularly neutrophils, as regulators of skeletal muscle metabolism. Neutrophils are rapidly recruited into muscle after contraction or mild mechanical stress, where they accumulate along the capillary network and establish localized 'immunometabolic niches', microenvironments in which immune activity and metabolic responsiveness are closely integrated (Chaweewannakorn et al., 2021; Tsuchiya et al., 2018). Their presence has been associated with local upregulation of myokines such as IL-6 and CXCL1, as well as enhanced GLUT4 translocation (Chaweewannakorn et al., 2020). We have recently shown that pharmacological blockade of CX3CR1 and CXCR2 suppresses both neutrophil recruitment and exercise-induced GLUT4 translocation, presumably via neutrophil–endothelial interactions (Nyasha et al., 2023), suggesting that intramuscular neutrophil accumulation is indispensable for normal metabolic responses to exercise. However it has remained unknown whether these neutrophils also contribute to the post-exercise increase in insulin sensitivity, which represents the hallmark metabolic adaptation to exercise.

Neutrophil extracellular traps (NETs) are web-like chromatin structures decorated with histones and neutrophil granule proteins. They are classically known to trap pathogens during infection (Brinkmann et al., 2004), and we previously demonstrated that they are also generated in skeletal muscle after high-intensity eccentric contractions that cause muscle damage, where they regulate sterile inflammation and tissue repair (Suzuki et al., 2022). Importantly recent studies have shown that even physiological exercise can induce transient NET release in the circulation (Beiter et al., 2015; Fridlich et al., 2023). However whether NETs are also formed locally within skeletal muscle during mild, non-damaging contractions such as physiological exercise, and whether they exert beneficial physiological roles rather than adverse actions, remain largely unknown.

To address these questions under well-controlled experimental conditions, we used a sciatic nerve-mediated electric pulse stimulation (EPS) *in situ* contraction model that reliably mimics key features of physiological muscle activity (Nyasha et al., 2025), in combination with GLUT4-EGFP transgenic mice (Hatakeyama & Kanzaki, 2017), *in vivo* visualization of neutrophils and extracellular DNA, enzymatic NET degradation and paired analysis of stimulated versus contralateral control muscles. We further assessed the impact on insulin signalling intermediates, glucose uptake and finally confirmed physiological relevance in a voluntary running model. These experiments were designed to determine whether neutrophils contribute to the post-exercise increase in insulin sensitivity, and whether NET formation provides a structural basis for this effect.

## Methods

### Animal models

Male C57BL/6J mice, at 6–8 weeks of age, weighing 23–26 g were purchased from CLEA-Japan (Tokyo, Japan). For analyses of GLUT4 translocation, skeletal muscle-specific myc-GLUT4-EGFP transgenic mice (Hatakeyama & Kanzaki, 2017) were also used. In the design and performance of these experiments, the principles of laboratory animal care (NIH Publication No. 86-23, revised 1985) were followed, with adherence to specific national laws. All experimental procedures were approved by the Ethics Committee for Animal Experiments of Tohoku University (approval numbers: 2020DNA001-01, 2020BeLMO-008-02, 2020BeLMO-006-02 and 2024BeLMO-006-01) and performed in accordance with institutional guidelines. Mice were housed under a 12:12 h light/dark cycle at 23 ± 1°C with free access to standard chow (Labo MR Standard, Nihon Nosan Kogyo, Japan) and water.

### *In situ* muscle contraction and multiphoton imaging

*In situ* muscle contraction experiments were performed in 12- to 15-week-old mice as described previously (Nyasha et al., 2025). After a 16-h fast mice were anaesthetized by intraperitoneal injection of a mixture consisting of medetomidine hydrochloride (ZENOAQ, Fukushima, Japan, 0.3 mg/kg), midazolam (SANDZ, Tokyo, Japan, 4.0 mg/kg) and butorphanol (Meiji Seika Pharma Co, Tokyo, Japan, 5.0 mg/kg), and the sciatic nerve was exposed. A subminiature platinum slide electrode (EKT-0810, Bioresearch Centre, Tokyo, Japan) was placed on the nerve and connected to a stimulator (STG4004, Multi-Channel Systems, Reutlingen, Germany) to evoke unilateral hind-limb contractions for 3 min (train rate, 1 s; train duration, 1 s; pulse rate, 50 Hz; duration, 1 ms at 2.5 mA). The contralateral leg served as a sham-operated control. In some experiments QD655-conjugated anti-Gr-1 antibody (3 μg in 150 μl saline) was injected via the tail vein immediately after EPS, and muscles were fixed 1 h later for

quantification of neutrophils (Chaweewannakorn et al., 2020) (Fig. 1*A*). For the assessment of post-contraction insulin sensitivity (Fig. 3*A*), mice were maintained under anaesthesia for 2 h after EPS, then low (∼0.17 nM) or high (∼17 nM) doses of insulin were injected intravenously, and extensor digitorum longus (EDL) muscles were harvested ∼30 min later. Isoflurane (Mylan Pharmaceuticals, Tokyo, Japan) was supplied as needed to maintain anaesthesia. In some experiments 2-deoxyglucose (2-DG; 3 mmol/kg) was co-injected with insulin, and EDL muscles were collected 20 min later for glucose uptake measurements (Nyasha et al., 2025).

To visualize the capillary network and assess potential vascular and sarcolemmal disruption, we evaluated Evans Blue Dye (EBD) extravasation, which is widely regarded as one of the most reliable *in vivo* methods for detecting muscle fibre membrane disruption and vascular leakage (Hamer et al., 2002; Straub et al., 1997). EBD was intravenously administered immediately before mice were killed, followed by perfusion fixation and two-photon imaging.

All experiments were performed under continuous anaesthesia, and both hindlimbs were exposed to the same systemic anaesthetic conditions. Electrical stimulation was applied only to the sciatic nerve of one limb, whereas the contralateral limb served as an internal control. This unilateral design enables within-animal comparisons of contraction-induced responses, thereby minimizing potential confounding effects of anaesthesia immune cell recruitment and insulin action.

For experiments assessing post-exercise insulin responsiveness, QD655-conjugated anti-GR1 antibody was administered 1 h after EPS and allowed to circulate for 1 h before insulin stimulation, a timing chosen to label neutrophils previously recruited to exercised muscle while minimizing prolonged systemic GR1 antibody exposure and associated neutropenia.

Multiphoton imaging was performed after perfusion fixation with 4% paraformaldehyde in PBS. To visualize neutrophils or NETs, QD655-anti-Gr-1 (1.5 µg per mouse) or Sytox Orange (3 µl in 150 µl saline) (S11368, Thermo Fisher Scientific, Eugene, OR, USA) were injected 1 h after EPS and 1 h before insulin. To visualize the capillary network 2% Evans blue (056-04061, FUJIFILM Wako Pure Chemical Corporation, Osaka, Japan) (150 µl) was injected via the tail vein immediately before the mice were killed to minimize potential non-specific inhibition of cellular signalling, including purinergic receptor-mediated pathways (Burnstock, 2006; Yao et al., 2018). Dissected EDL muscles were imaged using an A1R-MP multiphoton microscope (Nikon, Tokyo, Japan) equipped with GaAsP non-descanned detectors and a 25× water-immersion objective (NA 1.1). For analysis fluorescence intensity profiles across the sarcolemma were obtained using Fiji ImageJ as described in our pre-

vious report (Tsuchiya et al., 2018). For quantification of GLUT4-EGFP on the sarcolemma, square regions of interest (ROIs) were manually drawn across the sarcolemmal membrane. Fluorescence intensity profiles were extracted, and mean intensities within the sarcolemmal region were calculated. The same analysis was performed separately for neutrophil-positive areas, Sytox-positive (NETs-containing) areas and corresponding negative regions for comparison.

## Spatial quantification of GLUT4 relative to NETs

Three-dimensional confocal image stacks of skeletal muscle fibres (typical volume: 60 × 60 × 50 µm; $X \times Y \times Z$) were acquired to visualize GLUT4-EGFP and Sytox Orange-positive NET signals. Given that skeletal muscle fibres in our datasets typically exhibited diameters of ∼40–70 µm, spatial analysis along the $Y$-axis was restricted to a 30-µm window to capture a single sarcolemmal surface and avoid conflation of signals originating from opposing membrane sides. Because NET-associated GLUT4 enrichment appears as elongated, capillary-aligned domains rather than uniform sarcolemmal distributions, an axis-aligned projection strategy was used to preserve spatial continuity along the capillary axis while minimizing signal dilution from surrounding regions. The analysis was performed using the following standardized sequence:

(1) **Preprocessing**: Background fluorescence was subtracted from each channel (GLUT4-EGFP and Sytox) using identical settings across samples.
(2) **Axis alignment**: Image stacks were rotated so that the capillary-associated signal was oriented horizontally in the XY view.
(3) **$X$-range selection and $Z$–$Y$ projection**: A 30-µm segment along the $X$-axis containing the NET-positive (or NET-negative) region of interest was selected, and fluorescence within this segment was summed to generate a $Z$–$Y$ projection using a summation-based projection approach (Bolte & Cordelières, 2006).
(4) **ROI definition**: Within the resulting Z–Y projection, a 30-µm region along the Y dimension corresponding to the capillary-associated area was defined.
(5) **Y-collapse to generate a one-dimensional profile**: Fluorescence within the defined Y-range was summed to produce a one-dimensional intensity profile along the Z-axis, in which each point represents the summed signal across the selected X- and Y-ranges (Bolte & Cordelières, 2006). Because membrane-associated fluorescence intensity is substantially higher than that of intracellular regions, summation-based axis compression preserves membrane signals while reducing spatial

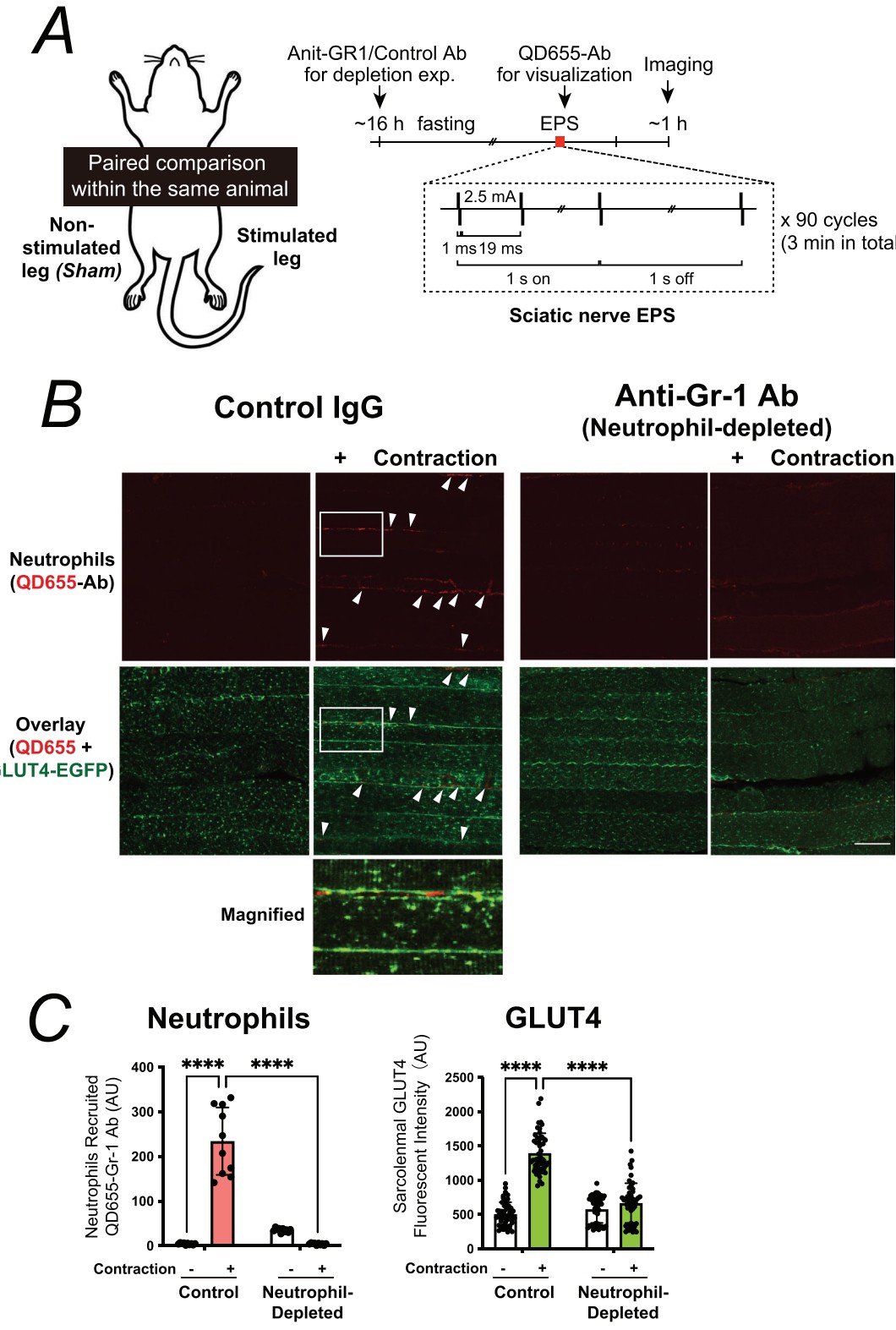

**Figure 1. Neutrophil depletion attenuates contraction-induced GLUT4 translocation in the sciatic nerve electric pulse stimulation (EPS) model**

*A*, experimental design. GLUT4-EGFP transgenic mice underwent unilateral sciatic nerve electrical stimulation for 3 min (train rate, 1 s; train duration, 1 s; pulse rate, 50 Hz; duration, 1 ms at 2.5 mA) after overnight fasting. Anti-Gr1 or control IgG was administered beforehand to deplete circulating neutrophils. QD655-conjugated anti-Gr1 antibody was injected to visualize neutrophils *in vivo* immediately before EPS. *B*, representative confocal images of

noise, enabling robust quantification of localized GLUT4 enrichment along the sarcolemma as a one-dimensional profile.

(6) **Quantification**: The area under the curve (AUC) of the GLUT4 intensity profile was calculated and compared between NET-positive and NET-negative regions. All image processing and quantitative analyses were performed using ImageJ/Fiji (Schindelin et al., 2012).

The workflow relies exclusively on standard ImageJ/Fiji operations (background subtraction, rotation and summation-based projections) and is fully reproduced from the step-by-step procedure described above. This workflow enables the sensitive detection of spatially restricted GLUT4 enrichment along NET-rich, capillary-aligned regions that would be underestimated by conventional two-dimensional projections or whole-fibre averaging.

### Wheel running model

A forced running model (Nyasha et al., 2023) was used to examine physiological exercise-induced NET formation. Briefly after 2 h without access to food, each mouse (12–15 weeks old) was placed in a cylindrical cage (17 cm diameter, 7 cm width) with stainless steel rods (2 mm diameter, 1cm apart) (Lafayette Instrument, Model 80801, Lafayette, IN, USA). The cage was then rotated with an electric motor at a walking speed of ∼12 m/min (Lafayette Instrument, Models 35500 and 80800A) at ∼12 m/min for 2 h at room temperature (23 ± 1°C). For cfDNA visualization, Sytox Orange was intravenously injected immediately before running session. After completion of the running protocol, mice were killed and immediately subjected to perfusion fixation, after which skeletal muscles were harvested and processed for imaging as described above.

### Neutrophil depletion and pharmacological inhibition

For neutrophil depletion anti-Gr-1 antibody (RB6-8C5; 0.5 μg/g body weight) was injected intravenously 12 h before EPS, as described previously (Tsuchiya et al., 2018), a protocol known to induce neutropenia for several days (Chen et al., 2001). Control mice received equivalent amounts of normal rat IgG (Code No. 147-0 9521, Wako, Japan). To suppress neutrophil recruitment pharmacologically, AZD8797 (21.8 μg; CX3CR1 antagonist) (No. 2255; Axon Medchem, Reston, VA, USA) and SB225002 (50 μg; CXCR2 antagonist) (No. 0000088761; Sigma Aldrich, St. Louis, MO, USA) were co-injected intravenously 15 min before EPS as we previously reported (Nyasha et al., 2023). To degrade NETs, DNase I (200 μg in 150 μl saline) (Sigma-Aldrich) was injected intravenously 1 h after EPS or before running.

### Glucose uptake assay

Glucose uptake was measured using 2-DG as described previously (Nyasha et al., 2025). Two hours after EPS, 2-DG (3 mmol/kg body weight) with or without insulin was injected via the tail vein. After 20 min, EDL muscles were homogenized in PBS, and the homogenates were centrifuged at 1000 *g* for 10 min at 4C, and the supernatants were assayed for glucose uptake using the Glucose Uptake-Glo assay (Promega, Madison, WI, USA), which selectively detects intracellular 2-DG-6-phosphate accumulation and does not measure free 2-DG or endogenous glucose-6-phosphate (Durumutla et al., 2024; Ueyama et al., 2000). Luminescence was recorded using a SpectraMax M5-LV Multi-Mode Microplate Reader (Molecular Devices, Tokyo, Japan), and data were analysed based on a 2-DG-6-phosphate standard curve to ensure quantification accuracy.

### Molecular analyses

For qRT-PCR total RNA was isolated using TRI reagent, reverse-transcribed using the Transcriptor First Strand cDNA Synthesis Kit (Roche) and analysed on Bio-Rad CFX Connect Systems (Bio-Rad Laboratories Inc., Hercules, CA, USA) with a SsoAdvanced Universal SYBR Green Supermix. Relative expression levels were normalized to 36B. The primer sequences were as follows:

| | |
|---|---|
| Mouse IL-6 | Forward 5′-CAATGCTCTCCTAACAGATAAG-3′ |
| | Reverse 5′-AGGCATAACGCACTAGGT-3′ |
| Mouse CXCL1 | Forward 5′-GCTGGCTTCTGACAACACTAT-3′ |
| | Reverse 5′-CAAGCAGAACTGAACTACCAT-3′ |
| Mouse Ly6G | Forward 5′-ACTCTGGACAATACTGAGATCACT-3′ |
| | Reverse 5′-GGTCTGCAGAAGGACTGAAAC-3′ |
| Mouse 36B | Forward 5′-CGACCTGGAAGTCCAACTAC-3′ |
| | Reverse 5′-ATCTGCTGCATCTGCTTG-3′ |

For immunoblotting muscle lysates were prepared in lysis buffer, separated by SDS-PAGE and probed with specific antibodies against total Akt, phospho-Akt (Ser473), phospho-AS160 (Thr642), phospho-AS160 (Ser704) and phospho-TBC1D1 (Ser237), followed by chemiluminescent detection (SuperSignal West Femto, Thermo Fisher, Carlsbad, CA, USA). Total protein concentrations were measured using a BCA Protein Assay Kit (Thermo Fisher Scientific). Anti-phospho-TBC1D1 (Ser237) and anti-phospho-AS160 (Ser704) antibodies were generated by immunizing rabbits with keyhole limpet haemocyanin-conjugated peptides (H-CRPMRKSF-pS-QPGLRS-OH and H-TSF-pS-APSFTA-OH, respectively), and purified by immunoaffinity chromatography using SulfoLink Coupling Gel (Pierce) (Hatakeyama et al., 2019). Commercial antibodies were obtained as follows: anti-AS160 (Sigma, SAB4200101), anti-phospho-AS160 (Thr642) (Cell Signaling Technology, #8881), anti-Akt (Cell Signaling Technology, #9272) and anti-phospho-Akt (Ser473) (Cell Signaling Technology, #9271). These commercial antibodies were used at 1:2000 dilution for immunoblotting, whereas the laboratory-generated antibodies were used at a final concentration of 0.5 µg/mL. For detection, HRP-conjugated secondary antibodies were used at a dilution of 1:20,000, including goat anti-rabbit IgG (H+L) (Thermo Fischer, #32460) and goat anti-mouse IgG (H+L) (Thermo Fischer, #32430). For the ELISA assay the whole EDL muscle tissue lysates extracted were adjusted to 1 mg/mL with lysis buffer. Myeloperoxidase (MPO) levels were quantified using DuoSet ELISA kits (detection range 25–1600 pg/mL), (#DY3667, R&D Systems, Minneapolis, MN, USA).

### Statistical analysis

Statistical analyses were performed using GraphPad Prism 9 (GraphPad Software, San Diego, CA, USA). Data are presented as means ± SD unless otherwise specified. Group comparisons were conducted using unpaired Student's *t* tests or multifactorial ANOVA followed by Tukey's *post hoc* test. A *P*-value < 0.05 was considered statistically significant.

## Results

### Neutrophil recruitment is induced by sciatic nerve-mediated contraction and contributes to GLUT4 translocation in skeletal muscle

The contribution of neutrophils to exercise-induced GLUT4 translocation has been demonstrated in several of our previous studies (Chaweewannakorn et al., 2020; Chiba et al., 2015; Tsuchiya et al., 2018) and further supported by pharmacological inhibition experiments (Nyasha et al., 2023).

To detect subtle contraction-induced differences with greater precision and to accurately assess post-exercise increases in insulin sensitivity within the same experimental framework, we established a paired experimental model using sciatic nerve-mediated electrical stimulation (EPS) in GLUT4-EGFP transgenic mice (Fig. 1A). In this model, unilateral muscle contraction was induced by EPS, whereas the contralateral leg served as a non-stimulated control, allowing within-animal comparison under identical systemic conditions (Nyasha et al., 2025). Using this model we first examined whether neutrophil recruitment contributes to contraction-induced GLUT4 translocation.

We first confirmed that this paired EPS model faithfully recapitulates exercise-induced neutrophil recruitment and GLUT4 translocation. In control IgG–treated GLUT4-EGFP mice, robust recruitment of QD655-labelled neutrophils was detected in the interstitial space of contracted muscles, coinciding with marked sarcolemmal GLUT4-EGFP translocation (Fig. 1B). In contrast the depletion of circulating neutrophils by pretreatment with an anti-Gr1 antibody, as we previously reported (Tsuchiya et al., 2018), almost completely abolished contraction-induced neutrophil accumulation and markedly attenuated GLUT4 translocation to the sarcolemma (Fig. 1B and C). Furthermore pharmacological blockade of neutrophil recruitment by combined administration of CX3CR1 and CXCR2 antagonists (AZD + SB), also as described previously (Nyasha et al., 2023), reproduced this effect: AZD + SB markedly reduced neutrophil accumulation in response to contraction and strongly suppressed sarcolemmal GLUT4 translocation (Fig. 2A and B). Spatial analysis further revealed that, in contracted EDL under low-insulin conditions, local enrichment of sarcolemmal GLUT4-EGFP signal intensity was significantly higher in regions adjacent to recruited QD655-labelled neutrophils, compared to neutrophil-free regions (Fig. 2B, *right*).

Furthermore contraction increased mRNA levels of neutrophil-associated genes (Ly6G), upregulated the intramuscular expression of the well-characterized contraction-responsive myokines CXCL1 and IL-6 (Farmawati et al., 2013; Nedachi et al., 2008; Nedachi et al., 2009), and elevated the local concentration of the neutrophil granule protein MPO. Importantly those contraction-induced responses, which we previously observed in a running model (Tsuchiya et al., 2018) and in a masseter gnawing exercise model (Chaweewannakorn et al., 2020), were faithfully reproduced in the present sciatic nerve EPS-induced contraction model. Moreover they were almost completely abolished by AZD + SB treatment (Fig. 2C).

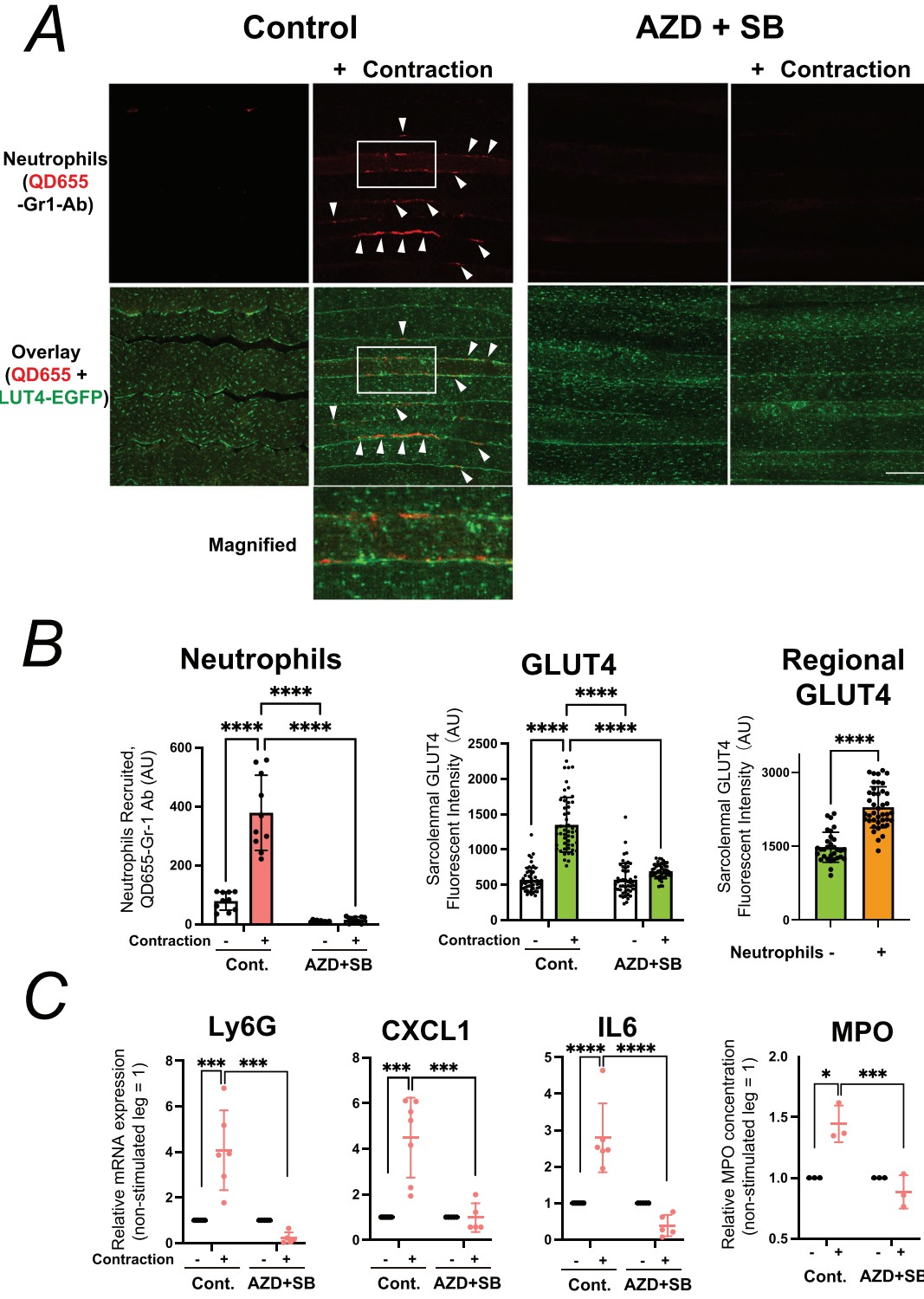

**Figure 2. Pharmacological blockade of neutrophil recruitment suppresses contraction-induced GLUT4 translocation in the sciatic nerve electric pulse stimulation (EPS) model**
*A*, representative confocal images of extensor digitorum longus (EDL) muscles from control or CX3CR1/CXCR2 antagonist (AZD + SB)–treated mice, showing neutrophils (QD655, *red*) and GLUT4-EGFP (*green*). Arrowheads mark recruited neutrophils and surrounding GLUT4 translocation. Insets: magnified view of boxed region. Three independent experiments were performed, and representative images are shown. Scale bar: 50 µm. *B*, quantification of recruited neutrophils (*left*), sarcolemmal GLUT4 intensity (*middle*) and regional GLUT4 enrichment

in the vicinity of neutrophils (*right*) (n = 4–9 mice per group). Data are mean ± SD; ****$P < 0.0001$ by two-way ANOVA. *C*, Contraction-induced increases in Ly6G mRNA and MPO protein (neutrophil markers) and CXCL1 and IL-6 mRNAs (muscle-derived myokines) mRNAs in whole EDL muscle lysates (n = 3–8 mice per group). All responses were markedly suppressed by AZD + SB treatment. Data are mean ± SD; *$P = 0.020$, ***$P < 0.001$, ****$P < 0.0001$.

Overall these findings indicate that sciatic nerve-mediated EPS-induced muscle contraction elicits coordinated responses encompassing neutrophil recruitment, local myokine induction and GLUT4 translocation to the sarcolemma, similar to the responses observed in skeletal muscle after *in vivo* exercise. Because this EPS model enables direct within-animal comparisons between stimulated and contralateral non-stimulated muscles, it provides a powerful and reproducible platform for dissecting the local immune-metabolic crosstalk underlying contraction-induced GLUT4 mobilization.

## Post-contraction neutrophil recruitment enhances local insulin sensitivity for GLUT4 translocation

In this EPS model acute contraction-induced stimulation of glucose uptake returns to basal levels within 2 h after EPS-evoked muscle contraction, enabling the specific evaluation of post-contraction increases in insulin responsiveness. As previously reported (Nyasha et al., 2025), low-dose insulin slightly increased 2-DG uptake in control muscles, whereas prior contraction significantly potentiated low-insulin-dependent 2-DG uptake; high-dose insulin stimulated glucose uptake similarly with or without prior contraction. To elucidate the cellular mechanisms underlying post-contraction enhancement of insulin responsiveness, particularly the involvement of neutrophils, we investigated whether neutrophil recruitment during the post-contraction period contributes to the enhancement of local insulin sensitivity. To address this question we used GLUT4-EGFP mice, which enabled high-resolution confocal visualization of insulin-stimulated GLUT4 translocation. The experimental procedure is illustrated in Fig. 3*A*: mice underwent unilateral sciatic nerve EPS, followed by a 2 h rest period and intravenous injection of low-dose insulin.

For visualization of neutrophils, as mentioned above, QD655-conjugated anti-GR-1 antibodies were intravenously administered via the tail vein 1 h before insulin injection. Approximately 25 min after insulin administration, the mice were perfusion-fixed, and EDL muscles were dissected for imaging. Very faint QD655-labelled GR-1-positive signals, likely representing residual neutrophil-derived material, were still detectable in contracted muscles (Fig. 3*B*), and their abundance was markedly reduced by CX3CR1/CXCR2 antagonist treatment (AZD + SB) (Fig. 3*C, left*). Low-dose insulin significantly increased sarcolemmal GLUT4 translocation

in contracted muscles, but this effect was largely abolished by AZD + SB treatment (Fig. 3*C, middle*). Furthermore spatial analysis revealed that, under post-contraction low-insulin conditions, sarcolemmal GLUT4 enrichment was higher in the vicinity of residual QD-positive remnants than in QD-poor regions (Fig. 3*C, right*). These results indicate that contraction-induced neutrophils, detected as residual Gr1-positive remnants, remain sparsely detectable during the post-contraction period and are associated with localized microenvironments that potentiate insulin-stimulated GLUT4 translocation.

To exclude the possibility that the observed effects were confounded by muscle injury under the present EPS conditions, we conducted an additional set of experiments using protocols largely similar to those shown in Figs 1*A* or 3*A*, differing only in the timing of EBD administration (Fig. 4). Specifically EBD was intravenously administered immediately before the mice were killed to avoid potential interference with purinergic signalling, followed by perfusion fixation and two-photon imaging of dissected EDL muscles. Under these conditions EBD was strictly confined to the vasculature, with no detectable extravasation into the muscle interstitium or myofibres, indicating that the sciatic nerve-mediated EPS contraction protocol did not induce vascular leakage or myofibre membrane damage (Fig. 4*A*). Furthermore high-resolution imaging revealed that, in previously contracted muscles under low-insulin conditions, sarcolemmal GLUT4-EGFP signals were preferentially enriched in regions adjacent to the capillary network (Fig. 4*B*), further supporting the crucial role of neutrophils within the immunometabolic niche in mediating the post-exercise, insulin-dependent, local enhancement of GLUT4 translocation.

## Extracellular traps formed after contraction contribute to local insulin-sensitizing effects on GLUT4 translocation

Because only faint residual GR-1-positive signals were detectable ∼2.5 h after contraction (Fig. 3), we examined whether the previously recruited neutrophils had undergone NET-like changes and left DNA-based remnants. To visualize cell-free DNA (cfDNA) *in vivo*, Sytox Orange was injected via the tail vein 1 h after EPS-induced contraction, and the mice were kept at rest for an additional hour. Subsequently low-dose insulin was intravenously administered, and muscles were harvested 25–30 min later for simultaneous analysis of GLUT4 trans-

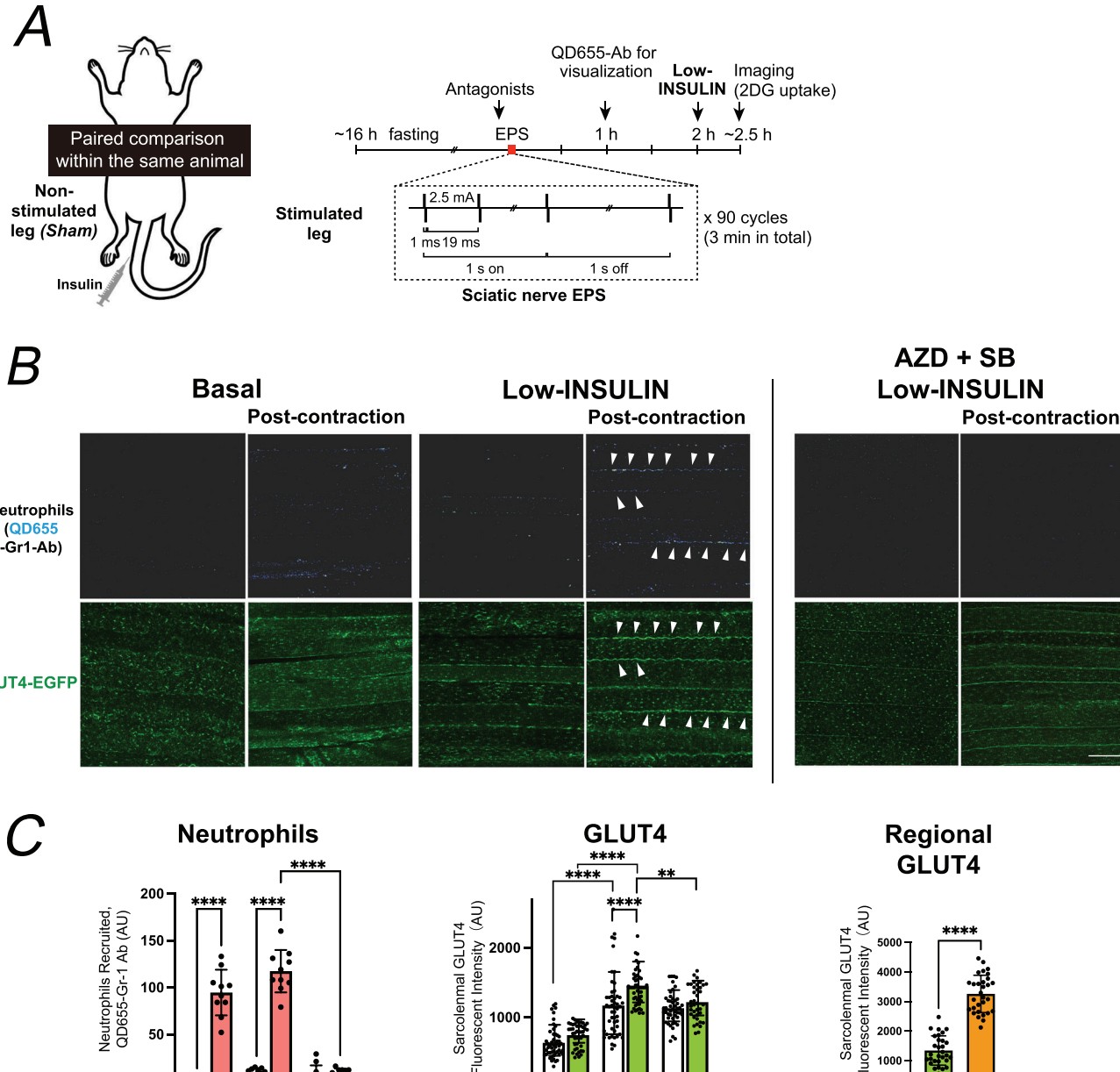

**Figure 3. Neutrophil recruitment is required for post-contraction enhancement of insulin sensitivity**
*A*, experimental design. GLUT4-EGFP mice underwent unilateral sciatic nerve electric pulse stimulation (EPS) for 3 min (train rate, 1 s; train duration, 1 s; pulse rate, 50 Hz; duration, 1 ms at 2.5 mA), followed by a 2-h rest period. Low-dose insulin was then injected intravenously, and GLUT4 translocation was analysed 25–30 min later. In most experiments, except for Evans blue, dyes for visualization of neutrophils/cfDNA were administered intravenously at 1 h after EPS. QD655–anti-GR1 antibody was administered 1 h after EPS and allowed to circulate for 1 h before perfusion–fixation. *B*, to visualize residual neutrophil-derived material, QD655–conjugated anti-GR-1 antibodies were intravenously injected 1 h before insulin administration. Approximately 25 min after insulin injection, mice were perfusion-fixed and extensor digitorum longus (EDL) muscles were subjected to two-photon imaging. Representative confocal images of QD655-labelled neutrophils (*white-blue pseudo-colour*) and GLUT4-EGFP (*green*) in post-contraction and contralateral control muscles under basal, low-insulin or AZD + SB + low-insulin conditions. QD signals were faint due to the ∼2.5 h interval after EPS and were pseudo-coloured to aid visualization. Three independent experiments were performed, and representative images are shown. *C*, quantification of recruited neutrophil-derived remnants (QD655 signals) (*left*), sarcolemmal GLUT4 fluorescence

(*middle*) and regional GLUT4 enrichment in the vicinity of residual neutrophil-derived remnants (QD655 signals) (*right*). Low-dose insulin promoted GLUT4 accumulation only in previously contracted muscles, and this effect was abrogated by AZD + SB (*n* = 4–9 mice per group). Data are mean ± SD; **$P$ = 0.0031, ****$P$ < 0.0001 by two-way ANOVA.

location and cfDNA accumulation (Fig. 5*A*). cfDNA signals were markedly increased in previously contracted muscles compared to contralateral controls, indicating the presence of extracellular trap-like structures.

We next tested whether these extracellular structures contribute to the post-contraction enhancement of insulin sensitivity. DNase treatment, co-applied with Sytox Orange 1 h after EPS, eliminated Sytox-detectable cfDNA and suppressed low-dose insulin-induced GLUT4 translocation, whereas DNase treatment alone had no effect (Fig. 5*A* and *B*). Furthermore 2-DG uptake was significantly increased by contraction plus low-dose insulin, and this potentiation was blunted by DNase pre-treatment, whereas high-dose insulin elicited robust 2-DG uptake irrespective of DNase (Fig. 5*C*). Importantly these cfDNA signals were completely absent in mice rendered neutropenic by anti-GR1 antibody treatment initiated the day before the experiment (Fig. 6), supporting that neutrophils are a major source of the cfDNA.

To clarify how NET degradation affects insulin signalling, we examined the phosphorylation status of key signalling proteins in muscles harvested ∼10 min after insulin injection (Fig. 7). Phosphorylation of AS160 at T642 was significantly increased by contraction combined with low-dose insulin, and this increase was modest but significantly attenuated by DNase treatment, whereas high-dose insulin-induced phosphorylation at this site was unaffected. Phosphorylation of Akt at S473 exhibited a modest but non-significant increase with low-dose insulin, without any difference between contraction and non-contraction conditions. By contrast, high-dose insulin elicited significant phosphorylation at this site, which was significantly attenuated by DNase treatment. Phosphorylation of AS160 at S704 and TBC1D1 at S237 – sites primarily responsive to contraction rather than insulin – was significantly increased in contracted muscles compared to contralateral controls, but DNase treatment had no detectable effect.

Overall these results indicate that post-contraction extracellular DNA selectively augments low-dose insulin signalling at AS160 (T642) and modulates high-dose insulin signalling at Akt (S473), whereas contraction-driven phosphorylation at AS160 (S704) and TBC1D1 (S237) proceeds independently of extracellular DNA. Collectively our data suggest that neutrophil-derived extracellular traps persist after contraction and establish a local microenvironment that enhances insulin-stimulated GLUT4 translocation, at least in part by modulating canonical insulin signalling pathways.

## Regional enrichment of GLUT4 translocation near NETs

Given that contraction-induced neutrophil recruitment (Fig. 2*B*) and their residual GR-1-positive remnants (Fig. 3*B*) were spatially associated with locally enhanced GLUT4 translocation, we next examined in more detail whether the formation of NETs contributes to this regional potentiation of insulin responsiveness. To this end, we analysed post-contraction muscles exposed to low-dose insulin. Three-dimensional (3D) confocal image stacks of skeletal muscle fibres (typical volume: 60 × 60 × 50 μm) were acquired, capturing fluorescence signals for GLUT4 (EGFP) and cfDNA-positive NETs (Sytox).

For spatial quantification, the 3D stacks were compressed into ZY-plane projections and further collapsed along the *Y*-axis to generate one-dimensional intensity line profiles, in which each pixel represents the summed fluorescence intensity along the corresponding plane (Fig. 8*A*). This axis-aligned transformation enabled selective interrogation of a single sarcolemmal surface adjacent to NETs, while minimizing contributions from opposing membrane regions. A schematic overview of this axis-aligned spatial quantification workflow, optimized for capillary-associated signals, is shown in Fig. 8*A*, and representative intensity profiles are shown in Fig. 8*B*.

Although overall sarcolemmal GLUT4 translocation was detectable after low-insulin stimulation in contracted muscles, it was markedly elevated in regions containing NETs compared with surrounding NET-negative regions. Area-under-the-curve (AUC) quantification of the intensity profiles confirmed this localized enrichment (Fig. 8*C*), demonstrating that the NET-associated GLUT4 accumulation is spatially restricted rather than reflecting a uniform increase across the muscle fibre. Together these findings support the notion that NETs create confined microdomains that potentiate insulin-stimulated GLUT4 translocation, representing a mechanism likely underpinning the post-exercise enhancement of insulin sensitivity in skeletal muscle.

## Exercise-induced NET formation contributes to GLUT4 translocation also in voluntary running muscles

Finally, to confirm that the phenomena identified using the sciatic nerve-mediated contraction model – most notably the previously unrecognized formation of NETs – are also observed under physiological exercise conditions, we analysed skeletal muscles from GLUT4-EGFP mice subjected to 2 h of forced running

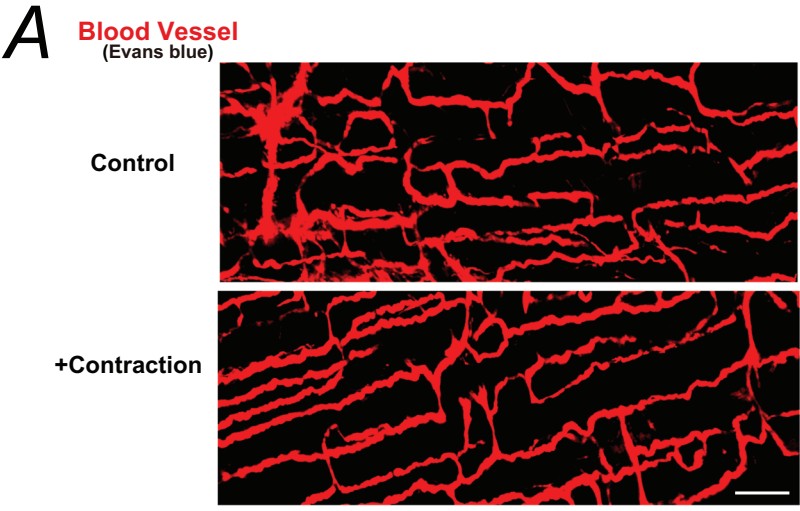

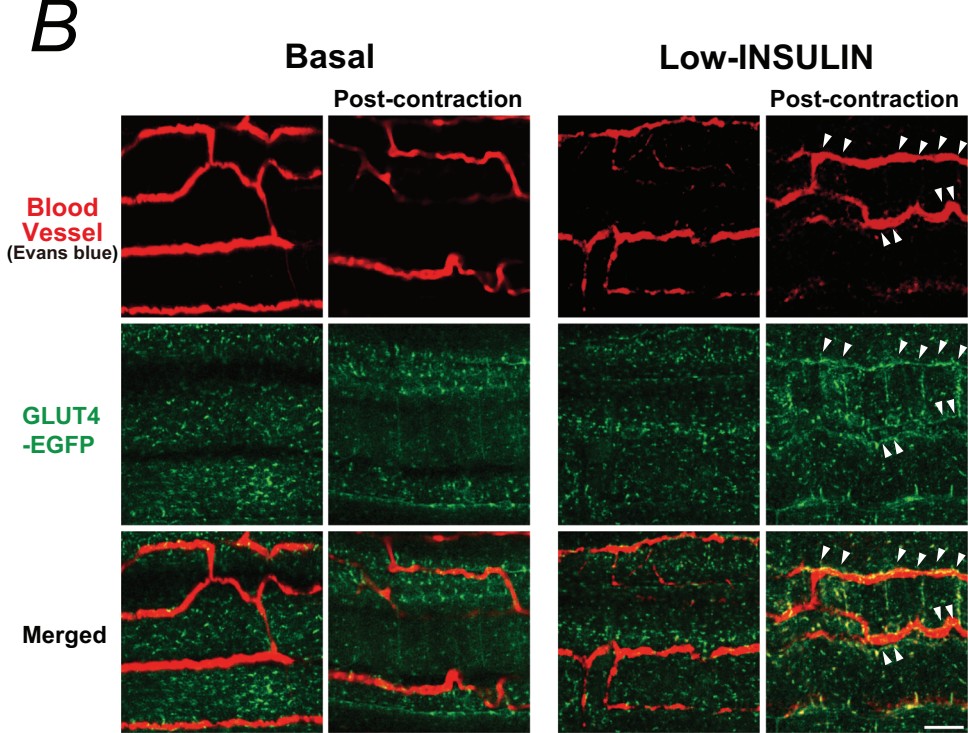

**Figure 4. Sciatic nerve-mediated contraction does not induce vascular leakage or myofibre membrane damage, but primes capillary-associated GLUT4 translocation in response to low-dose insulin**

*A*, to visualize the capillary network, Evans Blue Dye (EBD) was intravenously administered immediately before killing to avoid potential interference with purinergic signalling, followed by perfusion fixation and two-photon imaging of dissected extensor digitorum longus (EDL) muscles. Representative confocal images show EBD-labelled blood vessels *(red)* in muscles subjected to sciatic nerve–mediated electrical pulse stimulation (EPS)-induced contraction and contralateral control muscles. EBD was strictly confined to the vasculature, with no detectable extravasation into the muscle interstitium or myofibres, indicating that the contraction protocol did not induce vascular leakage or myofibre membrane damage. Scale bar: 50 μm. *B*, representative confocal images of EBD-labelled blood vessels *(red)* and GLUT4-EGFP *(green)* in post-contraction and contralateral control EDL muscles under basal or low-dose insulin conditions. Sarcolemmal GLUT4 translocation was observed exclusively in previously contracted muscles in response to low-dose insulin and was preferentially enriched along the capillary network *(arrowheads)*. Scale bar: 20 μm. Three independent experiments were performed, and representative images are shown.

(Tsuchiya et al., 2018) (Fig. 9*A*). Running markedly increased Sytox Orange-positive cfDNA structures in the muscle, indicating NETs are induced by mild exercise without detectable tissue damage (Fig. 9*B*). DNase treatment effectively eliminated these cfDNA signals and concomitantly reduced the running-induced sarcolemmal translocation of GLUT4 (Fig. 9*C*).

This demonstrates that the exercise-dependent formation of NETs, together with neutrophil recruitment, is required for the full induction of GLUT4 translocation during voluntary running. Because running reproduced the NET formation and DNase-sensitive GLUT4 translocation observed in the EPS model, these findings support the concept that neutrophil/NET-dependent local microenvironments broadly contribute to

exercise-induced enhancement of insulin responsiveness in skeletal muscle.

## Discussion

This study demonstrates that neutrophil recruitment and the subsequent formation of NETs create spatially confined immunometabolic niches along the capillary network. Such neutrophil-governed niches markedly potentiate local insulin sensitivity, as evidenced by enhanced GLUT4 translocation and glucose uptake, thereby providing a novel insight into the post-exercise increase in insulin sensitivity in skeletal muscle. This phenomenon has long been recognized (Cartee et al., 1989) and has been attributed mainly to multi-layered

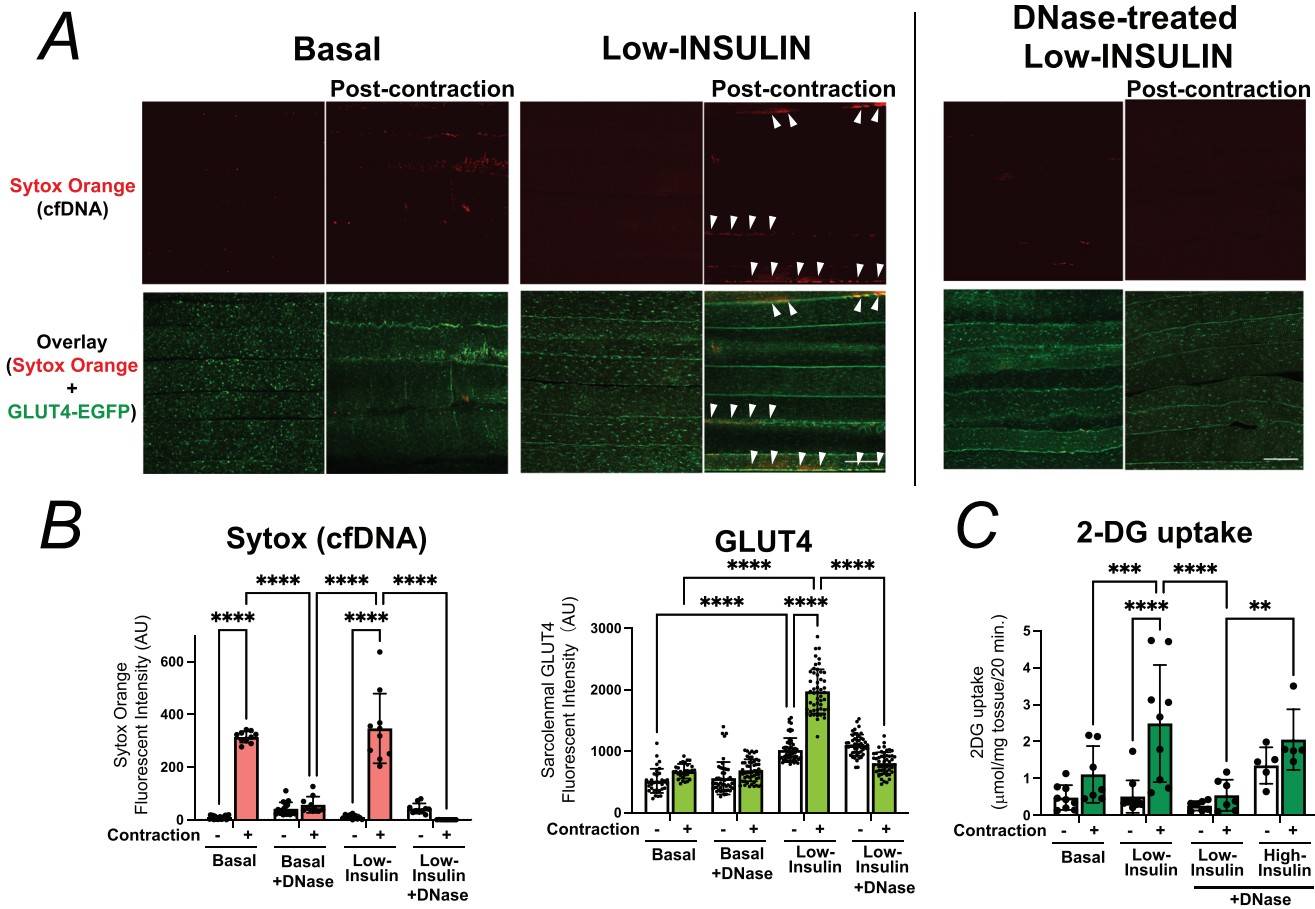

**Figure 5. Neutrophil-derived extracellular cell free DNA (cfDNA) contributes to post-contraction enhancement of insulin-stimulated GLUT4 translocation**

*A*, representative confocal images of Sytox Orange–labelled cfDNA (*red*) and GLUT4-EGFP (*green*) in post-contraction and contralateral control extensor digitorum longus (EDL) muscles under basal, low-insulin or DNase + low-insulin conditions. Three independent experiments were performed, and representative images are shown. *B*, quantification of Sytox Orange signals (*left*) and sarcolemmal GLUT4 fluorescence intensity (*right*) (*n* = 5–9 mice per group). DNase treatment reduced both cfDNA accumulation and GLUT4 translocation. *C*, 2-DG uptake assays showing that contraction enhanced low-dose insulin–stimulated glucose uptake, an effect abolished by DNase. High-dose insulin-stimulated uptake was unaffected (*n* = 5–9 mice per group). Data are mean ± SD; **$P$ = 0.0017, ***$P$ < 0.001, ****$P$ < 0.0001 by two-way ANOVA.

muscle-intrinsic and local tissue-level mechanisms, including sustained post-exercise phosphorylation of AS160/TBC1D4, enhanced GLUT4 trafficking competence and coordinated adaptations at the cellular and tissue levels (Funai et al., 2010; Hatakeyama et al., 2019; Hatakeyama & Kanzaki, 2017; Nedachi et al., 2008; Oki et al., 2018a; Oki et al., 2018b). However whether immune cells contribute to this phenomenon – one of the most crucial metabolic benefits conferred by exercise – has remained unclear.

Our present findings uncover a previously unrecognized role of neutrophil recruitment (Figs 1–3) and subsequent NET deposition (Figs 5, 6, 8 and 9) as a key local determinant of the well-known post-exercise increase in skeletal muscle insulin sensitivity. We previously showed that neutrophils promote local myokine expression and alleviate contraction-induced fatigue as well as enhance contraction-dependent GLUT4 translocation (Nyasha et al., 2023) (Chaweewannakorn et al., 2020; Tsuchiya et al., 2018). Consistent with this, pharmacological blockade of CX3CR1 and CXCR2 suppresses both neutrophil recruitment and exercise-induced GLUT4 translocation, presumably via neutrophil–endothelial interactions (Nyasha et al., 2023), indicating that intramuscular neutrophil accumulation is indispensable for normal metabolic responses to exercise. Of note, the CXCR2 ligands, including CXCL1, CXCL2 and CXCL5, are myokines secreted from contracting muscle cells (Chen et al., 2019), whereas the CX3CR1 ligand CX3CL1 is derived from endothelial cells, underscoring the importance of local intercellular crosstalk among muscle fibres, neighbouring endothelial cells and

recruited neutrophils in shaping an immunometabolic niche (Chaweewannakorn et al., 2020; Nyasha et al., 2023; Tsuchiya et al., 2018). The present study extends these observations by revealing that neutrophils, through the formation of NETs, are also indispensable for the endowment of heightened insulin sensitivity after exercise (Fig. 10), potentially in association with local myokine upregulation (Chen et al., 2019; Farmawati et al., 2013; Nedachi et al., 2009; Takahashi et al., 2022).

This represents a conceptual shift in understanding the role of neutrophil recruitment and NET formation in muscle tissue – from a damage-associated inflammatory response to a fine-tuned regulator of metabolic adaptation even during mild, non-damaging exercise. Although NETs are classically known to trap pathogens during infection (Brinkmann et al., 2004) and have been observed after intense eccentric muscle contractions (Suzuki et al., 2022), recent studies have shown that even physiological exercise can induce transient NET release in the circulation (Beiter et al., 2015; Fridlich et al., 2023). Our findings demonstrate that such NET-like cfDNA structures can also form locally within skeletal muscle after non-damaging contractions and mild running exercise (Figs 5, 6, 8 and 9), where they contribute to the regulation of local insulin responsiveness involving GLUT4 translocation. Thus in addition to serving as structural platforms for metabolic signalling, intramuscular NETs may facilitate improved glucose handling by acting in concert with exercise-induced increases in microvascular perfusion, together creating a permissive environment for efficient GLUT4-mediated glucose uptake.

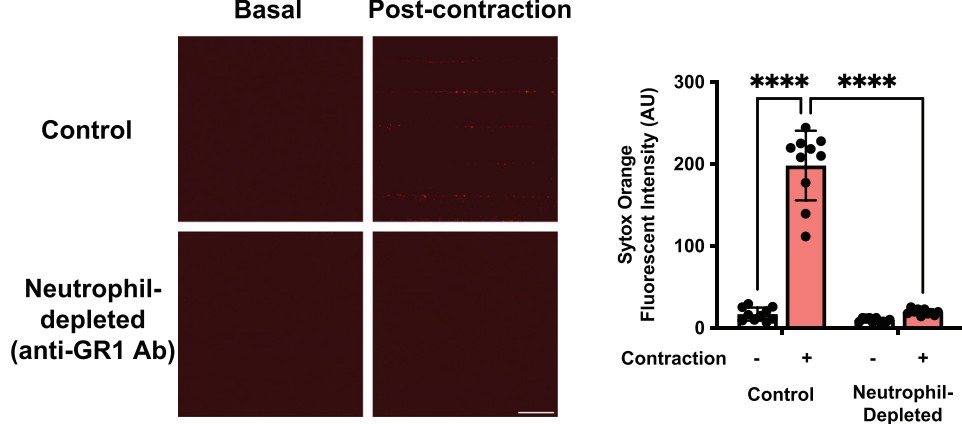

**Figure 6. Neutrophil depletion blunts contraction-induced neutrophil extracellular trap (NET) formation**
Intravascular Sytox Orange fluorescence (cell-free DNA; cfDNA) was assessed under basal conditions and following muscle contraction. In control mice, contraction markedly increased Sytox Orange signal, whereas this response was absent in mice rendered neutropenic by anti-GR1 antibody treatment initiated 1 day before the experiment, using the same depletion protocol as in Fig. 1*A*. Three independent experiments were performed, and representative images are shown. The graph on the right shows quantification of Sytox Orange fluorescence intensity (arbitrary units; AU; *n* = 4 per group). Data are presented as mean ± SD. Scale bar, 20 μm. ****$P$ < 0.0001.

Mechanistically NET-like cfDNA structures persisted for hours after contraction, especially along the capillary network (Figs 5 and 6), even after overt neutrophil signals had faded (Fig. 3). Sarcolemmal GLUT4 was locally enriched in these NET-positive regions, and DNase-mediated degradation of NETs eliminated cfDNA signals and attenuated the contraction-induced enhancement of GLUT4 translocation, glucose uptake (Fig. 5) and the potentiated phosphorylation of AS160 (T642) by low-dose insulin (∼0.17 nM) (Fig. 7). In contrast, when high-dose insulin (∼17 nM) was administered, 2-DG uptake was markedly increased regardless of DNase treatment (Fig. 5C), indicating that the contribution of NETs is particularly important under low, physiological insulin levels.

The modest yet significant reduction in AS160 (T642) phosphorylation with DNase treatment suggests that NETs may enhance local insulin accessibility, which could in turn facilitate stronger activation of insulin signalling and promote GLUT4 vesicle trafficking – an effect particularly evident under low-dose insulin stimulation. Disruption of extracellular DNA structures by DNase therefore supports the necessity of NET-like structures for enhanced insulin action at low, physiological insulin levels. However these data do not establish that NETs are sufficient on their own to potentiate insulin action. For example the present data cannot distinguish whether NETs directly potentiate insulin signalling at the myofibre surface or indirectly enhance insulin action by modulating local capillary insulin availability. Nevertheless the selective effect of DNase at low insulin doses, together with the spatial confinement of GLUT4 enrichment to NET-rich capillary regions, is more consistent with a model of local insulin potentiation near its physiological threshold rather than a global enhancement of insulin delivery.

In contrast this DNase-dependent effect was not observed at high insulin doses, where phosphorylation of AS160 (T642) (Fig. 7) and 2-DG uptake (Fig. 5C) were markedly increased irrespective of DNase treatment.

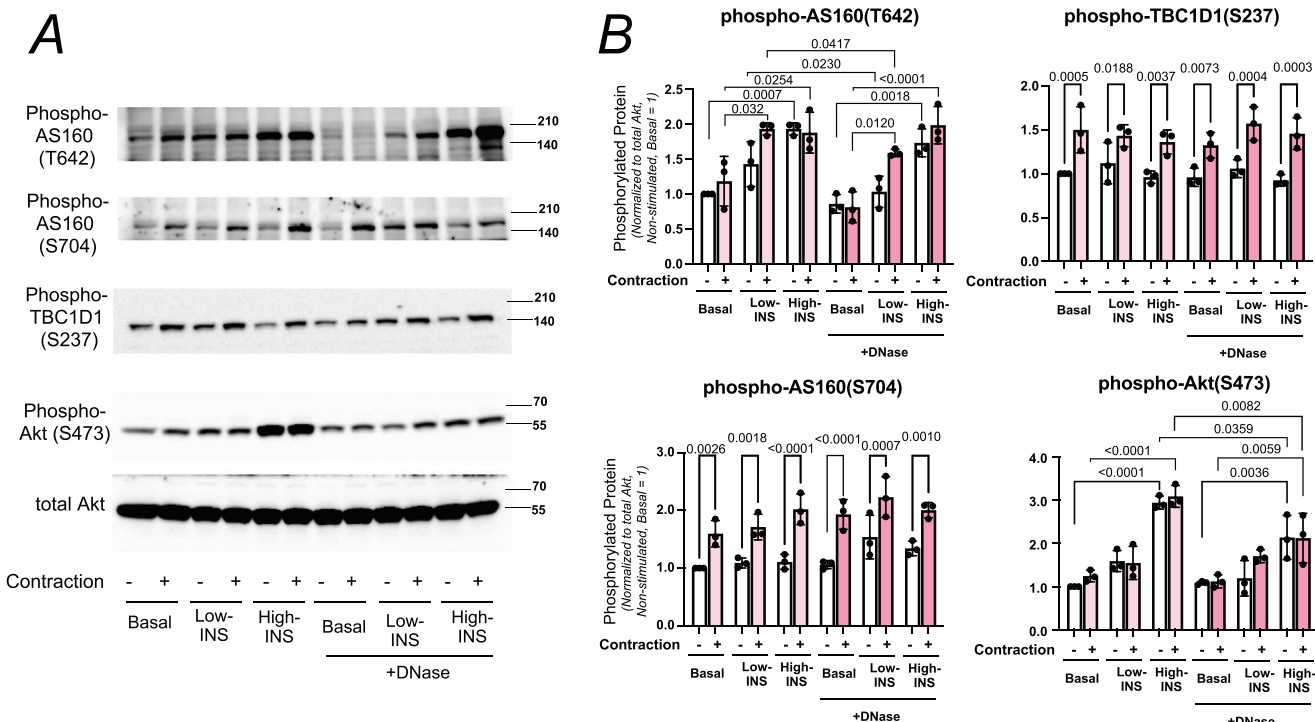

**Figure 7. DNase treatment attenuates low-dose insulin–induced phosphorylation of AS160 (T642) but not contraction-driven phosphorylation of AS160 (S704) or TBC1D1 (S237)**
*A*, representative immunoblots for phosphorylated and total forms of AS160, TBC1D1 and Akt in muscles under basal, low-insulin or high-insulin conditions, with or without DNase treatment. Three independent experiments were performed, and representative images are shown. *B*, quantification of phosphorylated protein levels, normalized to total Akt and expressed relative to contralateral non-stimulated muscles (set to 1) (*n* = 3 mice per group). Contraction combined with low-dose insulin significantly increased phosphorylation of AS160 (T642), which was modest but significantly reduced by DNase, whereas high-dose insulin–induced phosphorylation at this site was unaffected. Phosphorylation of Akt (S473) showed a slight increase with low-dose insulin and a mild, non-significant suppression by DNase. Phosphorylation of AS160 (S704) and TBC1D1 (S237) was significantly increased by contraction regardless of insulin and was not affected by DNase. Data are presented as mean ± SD. Statistical comparisons are indicated by connecting lines, with the corresponding *P*-values shown above each line.

Phosphorylation of Akt (S473) showed a slight but significant reduction with DNase treatment, particularly under high-insulin conditions, a pattern that differed from AS160 (T642) phosphorylation status (Fig. 7). Because AS160 and TBC1D1 are predominantly expressed in skeletal muscle fibres, their phosphorylation likely reflects myofibre responses, whereas Akt is also abundantly expressed in non-muscle cells such as vascular endothelial and smooth muscle cells. Thus caution is warranted when interpreting changes in Akt phosphorylation in whole-muscle samples.

Given that exercise is known to enhance vascular insulin sensitivity and increase capillary perfusion (Olver et al., 2019), and that NETs have been reported to transiently increase vascular permeability (Jorch & Kubes, 2017), it is plausible that NET deposition along the capillary network further augments local insulin access by modulating the perivascular microenvironment that constitutes these immunometabolic niches. Importantly this vascular-immune crosstalk aligns with evidence that acute exercise enhances insulin sensitivity through coordinated increases in microvascular perfusion and molecular signalling (Sjoberg et al., 2017). Nevertheless if increased insulin permeability were the sole mechanism, GLUT4 translocation would be expected to rise more uniformly across nearby muscle fibres; instead, the marked enrichment observed specifically at NET-rich sites (Fig. 8) suggests additional local regulatory effects, such as spatially confined facilitation of GLUT4 exocytosis and/or suppression of GLUT4 endocytosis. Overall these observations suggest that NETs help maintain a permissive perivascular environment that facilitates efficient insulin action and regulated GLUT4 trafficking, possibly through increased local vascular permeability as previously reported (Jorch & Kubes, 2017; Sjoberg et al., 2017).

The present findings position the NET-based immuno-metabolic niche as an integrative framework linking two well-established mechanisms underlying post-exercise insulin sensitization, involving both vascular and myofibre-intrinsic control of GLUT4 trafficking. Exercise promotes the localized formation of intravascular NETs within skeletal muscle, which may enhance or sustain microvascular permeability and insulin delivery beyond the immediate contractile period. In parallel, exercise also induces durable, myofibre-intrinsic changes in insulin responsiveness that potentiate the GLUT4 translocation machinery, as reflected by enhanced Akt-AS160/TBC1D4 and AMPK-TBC1D1 signalling. Within this framework the NET-rich perivascular niche may provide a spatially restricted context in which vascular and myofibre-intrinsic mechanisms converge to promote localized GLUT4 enrichment in myofibres, through mechanisms that remain to be elucidated.

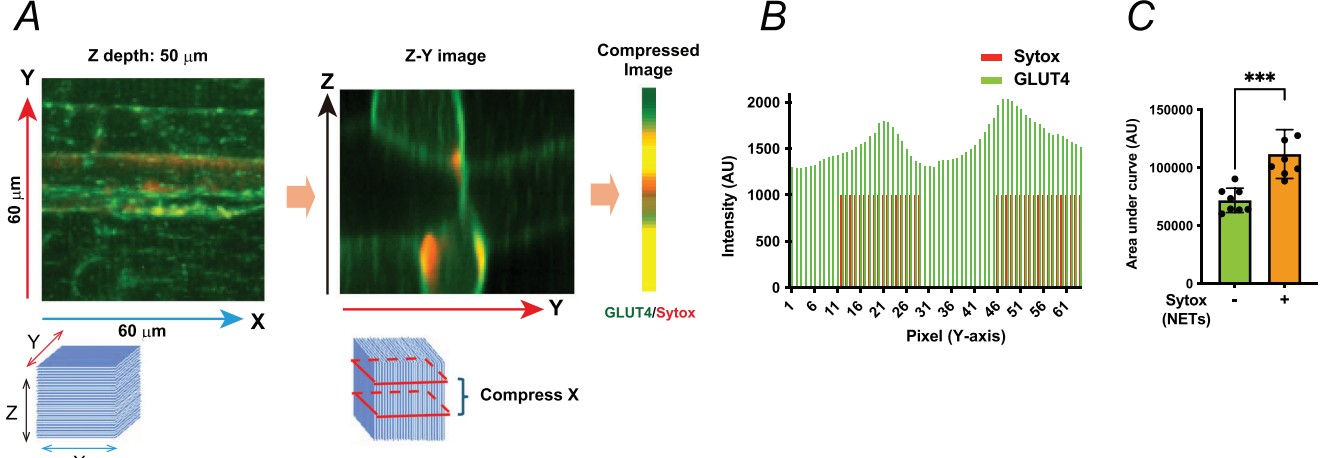

**Figure 8. Neutrophil extracellular trap (NET)-associated regional enhancement of low-insulin-induced GLUT4 translocation in post-contraction muscles**
*A*, representative confocal image stack of extensor digitorum longus (EDL) skeletal muscle fibres shown in the XY view (*top left*; Z depth = 50 μm) obtained after the completion of the electric pulse stimulation (EPS)-induced contraction protocol (post-contraction) and subsequent exposure to low-dose insulin (0.17 nM), as illustrated in Fig. 3*A*. The corresponding *Z–Y* image generated by summation along the *X*-axis (*top middle*) and the resulting one-dimensional compressed intensity image obtained by further summation along the *Y*-axis (*top right*) illustrate the axis-aligned spatial quantification of GLUT4 relative to NETs (see Methods for details). GLUT4-EGFP (*green*) and Sytox Orange-labelled cfDNA-positive NETs (*red*) are shown. *B*, pixel-wise fluorescence intensity profiles of GLUT4 (*green*) and Sytox (*red*) along the *Y*-axis from the compressed image shown in (*A*). *C*, area under the curve (AUC) analysis of GLUT4 fluorescence intensity in Sytox-positive (NET-rich) versus Sytox-negative (NET-poor) regions. This graph summarizes AUC values obtained by this method from pooled data of three independent experiments. Data are presented as mean ± SD. ****P* < 0.001.

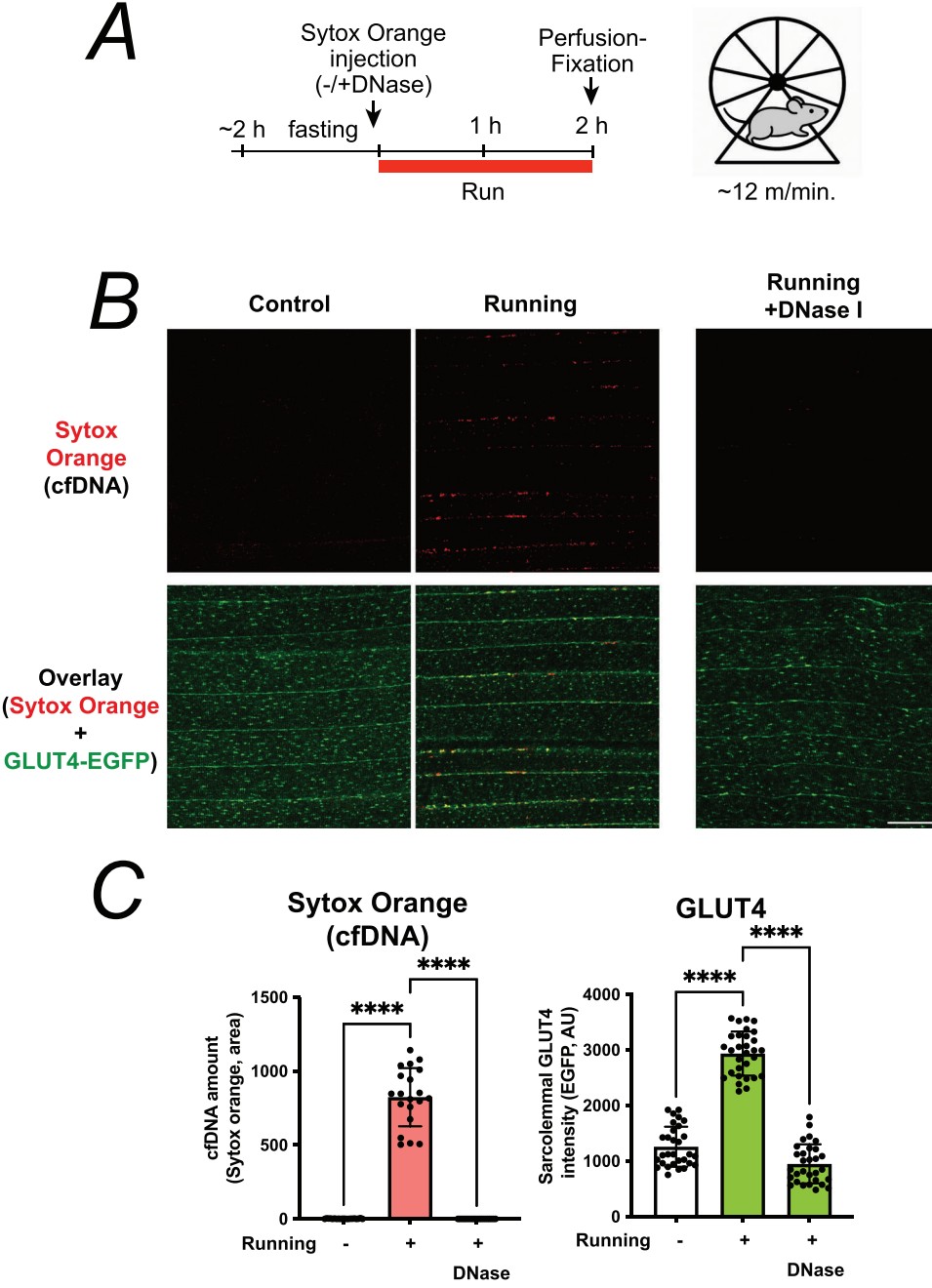

**Figure 9. Two hours of forced running induces neutrophil extracellular trap (NET) formation, which is required for exercise-dependent GLUT4 translocation**

*A*, experimental design. GLUT4-EGFP mice were subjected to running wheel exercise for 2 h. Sytox Orange and DNase I were administered via tail vein injection immediately before exercise. Following completion of the running protocol, mice were anaesthetized, perfusion-fixed and the extensor digitorum longus (EDL) muscle was analysed by two-photon microscopy. *B*, Representative confocal images of Sytox Orange-labelled cell-free DNA (cfDNA, *red*) and GLUT4-EGFP (*green*) in skeletal muscles from sedentary control mice, 2 h running mice and DNase-treated running mice. DNase or saline was administered via the tail vein immediately before the running session. Two hours of running induced prominent Sytox-positive cfDNA structures distributed along muscle fibres, indicating exercise-dependent formation of neutrophil extracellular traps (NETs). Three independent experiments were performed, and representative images are shown. Scale bar: 50 μm.

*C*, quantification of Sytox Orange-positive cfDNA area (*left*) and sarcolemmal GLUT4-EGFP fluorescence intensity (*right*). Running markedly increased both cfDNA accumulation and GLUT4 translocation, effects that were abolished by DNase treatment (*n* = 6 mice per group). Data are shown as mean ± SD; \*\*\*\**P* < 0.00001 by one-way ANOVA with Tukey's *post hoc* test.

NETs are not extracellular DNA scaffolds alone but also carry a variety of neutrophil-derived proteases, including elastase (NE), MPO, cathepsin G and proteinase 3 (PR3), which retain catalytic activity while bound to DNA (Korba-Mikolajczyk et al., 2025; Morales-Primo et al., 2022). In addition to remodelling extracellular matrix components, these enzymes can modulate endothelial basement membrane and glycocalyx structures, thereby influencing vascular barrier properties (Jerke et al., 2015; Manchanda et al., 2018). Moreover NET-associated remodelling of the perivascular extracellular matrix may influence integrin-mediated adhesion and signalling in endothelial cells and circulating leukocytes (Papayannopoulos, 2018), providing another potential mechanism for spatially restricted regulation at the muscle-vascular interface. Thus even NETs may establish specially confined immunometabolic niches capable of modulating the perivascular microenvironment without inducing diffuse tissue-level alterations.

Notably PR3 and NE can process and activate pro–IL-1 and other IL-1 family cytokines (Clancy et al., 2017). We previously demonstrated that neutrophil-derived IL-1 is essential for optimal muscle function, GLUT4 translocation and contraction-induced glucose uptake (Tsuchiya et al., 2018). Accordingly NET-associated proteases may contribute to a spatially regulated generation of IL-1 at the muscle-vascular interface, supporting beneficial signalling without excessive inflammation. Such spatial restriction may be particularly important under physiological insulin concentrations, where insulin signalling is modest and therefore more susceptible to local amplification.

Together NET-associated modulation of IL-1 signalling, perivascular extracellular matrix and endothelial function provides a plausible mechanism by which intravascular NETs enhance insulin responsiveness and GLUT4 translocation during and after muscle contraction. By locally amplifying insulin action along

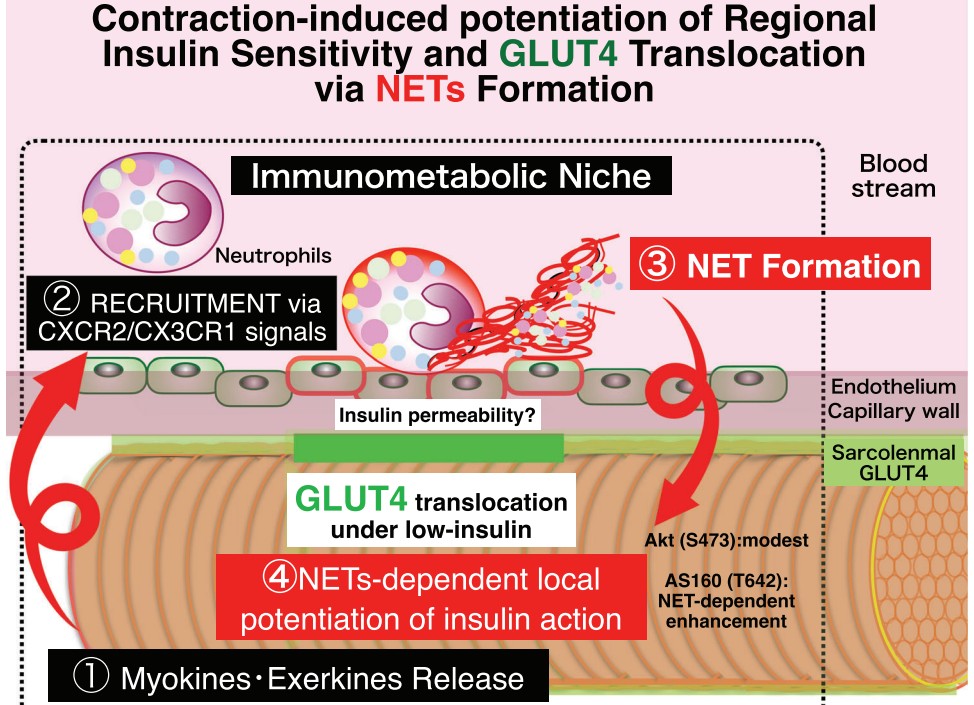

**Figure 10. Contraction-induced formation of a neutrophil extracellular trap (NET)-based immunometabolic niche potentiates regional insulin sensitivity and GLUT4 translocation**
Schematic model illustrating how skeletal muscle contraction promotes a NET-based immunometabolic niche that enhances regional insulin sensitivity and GLUT4 translocation under physiological, low-insulin conditions. Muscle contraction induces the release of myokines/exerkines (1), leading to neutrophil recruitment to the skeletal muscle microvasculature via CXCR2/CX3CR1 signalling (2). Recruited neutrophils form intravascular neutrophil extracellular traps (NETs) within the capillary network (3), which may increase local insulin permeability and modulate the perivascular microenvironment. NET formation locally potentiates insulin signalling in adjacent myofibres (4), resulting in enhanced Akt (S473) and AS160/TBC1D4 (T642) phosphorylation and promoting sarcolemmal GLUT4 accumulation despite low circulating insulin levels. Local GLUT4 accumulation may result from regional increased exocytosis and/or decreased endocytosis within specific regions. Together this spatially confined immunometabolic niche provides a mechanistic framework linking vascular and myofibre-intrinsic regulation of insulin action following muscle contraction.

the capillary network, these niches may promote efficient sarcolemmal GLUT4 accumulation in adjacent myofibres, where insulin and glucose are readily accessible. Importantly spatially confined amplification at the muscle-vascular interface provides a mechanistic basis for how a brief bout of exercise can enhance insulin efficiency at physiological insulin concentrations, despite minimal changes in systemic insulin availability. Future mechanistic studies will be required to elucidate the functional roles of regionally localized intravascular NET formation in driving spatially restricted biological effects within skeletal muscle following exercise.

Notably all experiments in this study were obtained from male mice using the electrically evoked contractions under anaesthesia in the predominantly fast-twitch EDL muscle, which has relatively sparse capillarization. Accordingly the scope of the present findings is currently limited to this experimental context. Because slow-twitch SOL muscle possesses a denser capillary network and exhibits higher basal insulin sensitivity (Holloszy, 2005), the extent to which a similar neutrophil/NET-dependent mechanism operates in slow-twitch muscle, as well as potential sex- or species-specific differences, remains unknown and should be considered hypothetical at this stage. It will therefore be important in future work to determine whether a similar neutrophil/NET-dependent mechanism also operates in other muscle types such as SOL muscle, potentially contributing to the well-known fibre type-specific differences in post-exercise insulin responsiveness (Pataky et al., 2019a; Pataky et al., 2019b).

To support the physiological relevance of this mechanism within the scope of the present study, complementary experiments using a 2-h voluntary running model demonstrated that running similarly induced prominent NET formation along the capillary network, and DNase treatment abolished both NET accumulation and running-induced GLUT4 translocation (Fig. 9). These findings indicate that NET formation is not restricted to electrically evoked contractions and occur under physiologically relevant exercise conditions; however its breadth across muscle fibre types and exercise modalities remains to be determined.

Given that exercise-induced insulin sensitization typically lasts for 24–48 h (Cartee et al., 1989), and that NETs are structurally stable and can persist in tissues for many hours – albeit mainly reported under conditions of intense, damage-inducing muscle contractions (Suzuki et al., 2022) – it is tempting to speculate that the persistence of NETs may contribute to the duration of this post-exercise insulin-sensitized state by maintaining perivascular immunometabolic niches, especially in highly capillarized muscle regions. This possibility remains hypothetical and should be directly tested in future studies.

In summary our findings establish that neutrophil recruitment and subsequent NET formation create spatially confined perivascular immunometabolic niches that underlie the post-exercise increase in insulin sensitivity, as manifested by enhanced insulin-stimulated GLUT4 translocation. Although the present study focused on acute post-exercise effects, this NET-based niche mechanism may undergo adaptive reinforcement and remodelling with repeated bouts of exercise. Such adaptations could contribute to the establishment and maintenance of robust insulin responsiveness, in concert with coordinated myokine expression profile, thereby supporting the chronic health benefits and structural adaptations of skeletal muscle. In contrast chronic low-grade inflammation associated with conditions such as obesity and insulin resistance may impair appropriate neutrophil recruitment, subsequent NET formation and the maintenance of these immunometabolic niches. Clarifying how chronic inflammation disrupts and how regular exercise restores these processes may provide new mechanistic insights into improving skeletal muscle glucose handling in insulin-resistant states.

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

## Additional information

### Data availability statement

All data supporting the findings of this study are included in the manuscript figures. No additional datasets were generated or analysed.

### Competing interests

The authors have no competing interests to declare.

### Author contributions

W.C. and M.K. conceived the project. M.K. supervised the study design and procedures. W.C. and M.K. performed all experiments. W.C. and M.K. analysed the data. M.K. wrote the manuscript.

### Funding

This work was supported in part by grants from the Japan Society for the Promotion of Science (no. 24K02874 to M.K.).

### Acknowledgements

We thank Natsumi Emoto for technical assistance.

### Keywords

exercise, GLUT4, immunometabolic niche, insulin sensitivity, NETs, neutrophil

## Supporting information

Additional supporting information can be found online in the Supporting Information section at the end of the HTML view of the article. Supporting information files available:

**Peer Review History**

