## [Peer Review History · The Journal of Physiology]

Intramuscular neutrophil-derived immunometabolic niches locally boost insulin-responsive GLUT4 translocation after muscle contraction

Weijian Chen and Makoto Kanzaki
DOI: 10.1113/JP290203

Corresponding author(s): Makoto Kanzaki (makoto.kanzaki.b1@tohoku.ac.jp)

Review Timeline:

Submission Date:	28-Sep-2025
Editorial Decision:	21-Oct-2025
Revision Received:	28-Dec-2025
Editorial Decision:	26-Jan-2026
Revision Received:	27-Jan-2026
Accepted:	03-Feb-2026

Senior Editor: Karyn Hamilton

Reviewing Editor: Nima Gharahdaghi

Transaction Report:

Re: JP-RP-2025-290203 "**Intramuscular neutrophil-derived immunometabolic niches locally boost insulin-responsive GLUT4 translocation after muscle contraction**" by Weijian Chen and Makoto Kanzaki

Dear Dr Kanzaki,

Thank you for submitting your manuscript to The Journal of Physiology. It has been assessed by a Reviewing Editor and by 2 expert referees and we are pleased to tell you that it is potentially acceptable for publication following satisfactory major revision.

Please address all the points raised and incorporate all requested revisions or explain in your Response to Referees why a change has not been made. We hope you will find the comments helpful and that you will be able to return your revised manuscript within 2 months. If your article is NOT for a Special Issue, you may have 9 months to revise. If you require an extension, please contact journal staff: jp@physoc.org. Please note that this letter does not constitute a guarantee for acceptance of your revised manuscript.

REVISION CHECKLIST:

We look forward to receiving your revised submission.

Yours sincerely,

Karyn Hamilton
Senior Editor
The Journal of Physiology

REQUIRED ITEMS

- The Journal of Physiology funds authors of provisionally accepted papers to use the premium BioRender site to create high resolution schematic figures. Follow this link and enter your details and the manuscript number to create and download figures. Upload these as the figure files for your revised submission. If you choose not to take up this offer, we require figures to be of similar quality and resolution. If you are opting out of this service to authors, state this in the Comments section on the Detailed Information page of the submission form. The link provided should only be used for the purposes of this submission. Authors will be charged for figures created on this premium BioRender account if they are not related to this manuscript submission.

- You must upload original, uncropped western blot/gel images (including controls) if they are not included in the manuscript. This is to confirm that no inappropriate, unethical or misleading image manipulation has occurred. These should be uploaded as 'Supporting information for review process only'. Please label/highlight the original gels so that we can clearly see which sections/lanes have been used in the manuscript figures. For more information, see: <https://physoc.onlinelibrary.wiley.com/hub/journal-policies#imagmanip>.

- Please ensure that the Article File you upload is a Word file.

- Papers must comply with the Statistics Policy: https://jp.msubmit.net/cgi-bin/main.plex?form_type=display_requirements#statistics.

In summary:

- If $n \leq 30$, all data points must be plotted in the figure in a way that reveals their range and distribution. A bar graph with data points overlaid, a box and whisker plot or a violin plot (preferably with data points included) are acceptable formats.

- If $n > 30$, then the entire raw dataset must be made available either as supporting information, or hosted on a not-for-profit repository, e.g. FigShare, with access details provided in the manuscript.

- 'n' clearly defined (e.g. x cells from y slices in z animals) in the Methods. Authors should be mindful of pseudoreplication.

- All relevant 'n' values must be clearly stated in the main text, figures and tables.

- The most appropriate summary statistic (e.g. mean or median and standard deviation) must be used. Standard Error of the Mean (SEM) alone is not permitted.

- Exact p values must be stated. Authors must not use 'greater than' or 'less than'. Exact p values must be stated to three

significant figures even when 'no statistical significance' is claimed.

- Please include an Abstract Figure Legend text within the main article file. The Abstract Figure is a piece of artwork designed to give readers an immediate understanding of the research and should summarise the main conclusions. If possible, the image should be easily 'readable' from left to right or top to bottom. It should show the physiological relevance of the manuscript so readers can assess the importance and content of its findings. Abstract Figures should not merely recapitulate other figures in the manuscript. Please try to keep the diagram as simple as possible and without superfluous information that may distract from the main conclusion(s). Abstract Figures must be provided by authors no later than the revised manuscript stage and should be uploaded as a separate file during online submission labelled as File Type 'Abstract Figure'. Please also ensure that you include the figure legend in the main article file. All Abstract Figures should be created using BioRender. Authors should use The Journal's premium BioRender account to export high-resolution images. Details on how to use and access the premium account are included as part of this email.

EDITOR COMMENTS

Reviewing Editor:

Comments to the Author:

This manuscript advances a clear conceptual shift: post-exercise insulin sensitization in skeletal muscle is proposed to be spatially gated by neutrophil-derived, NET-like extracellular DNA scaffolds laid down along the capillary meshwork, creating perivascular "immunometabolic niches" where physiological insulin preferentially drives GLUT4 translocation and glucose uptake. The claim is supported by paired in situ contraction and running models, two-photon/confocal imaging that localize GLUT4 enrichment to capillary-adjacent cfDNA structures, and DNase experiments that blunt low-insulin responses and attenuate AS160 (T642) phosphorylation, while sparing high-insulin effects-together positioning NET-rich perivascular topology as a structural determinant of insulin action after exercise.

Your DNase data convincingly show the necessity of extracellular DNA structures for low-dose insulin potentiation (GLUT4, 2-DG, AS160 T642), but do not establish sufficiency or exclude vascular delivery contributions. Could you please expand this point further?

- It seems it is necessary to have an integrative paragraph mapping the NET-niche model onto two established axes: (i) microvascular insulin delivery/perfusion and (ii) GLUT4 trafficking control (AS160/TBC1D1).

- All primary data are from male mice, fast-twitch EDL, anesthetized EPS, with a helpful validation in a running model. Please consider explaining these boundaries and broader claims (sex, species, slow-twitch SOL) as hypotheses.

- Your signature strength is spatial quantification (ZY compression → line-profile AUC). This editor suggests promoting this workflow from legend-level detail to the main text and stating clearly why this transformation is appropriate for capillary-aligned signals. Provide the analysis script, probably as supplemental, as this will materially increase reader confidence and citations.

- I suggest adding a central schematic that aligns time and space: contraction → neutrophil recruitment → NET deposition along capillaries → low insulin → localized GLUT4 translocation; include the differential effects at AS160 (T642) vs. Akt (S473). This will let general readers "see" the mechanism in 10 seconds.

Please consider clearly responding to this question in the discussion section: Why does a spatial, immune-governed mechanism matter for how one bout of exercise improves insulin efficiency at physiological insulin? Which could link to the translational relevance.

Senior Editor:

Comments for Authors to ensure the paper complies with the Statistics Policy:

If you choose to submit a revised manuscript for continued consideration, please visit the statistics policy to ensure compliance. The two main points that require revision are 1. reporting variability as SD rather than SEM and 2. reporting precise p-values. Please do note that it is acceptable for p values less than 0.001, to simply report $p < 0.001$. All others need precise p-value reporting. Thank you.

Comments to the Author:

Thank you for submitting your manuscript for consideration by The Journal of Physiology. As part of the peer review process, we recruited two Referees with expertise in this field of study. Both Referees were quite complimentary about aspects of the manuscript and they believe that, with adequate revision, the work has the potential to be influential. Therefore, at this point, we would like to invite you to respond point-by-point to each comment by the Referees and the

Reviewing Editor, making corresponding revisions to the manuscript. Please also ensure compliance with The Journal's statistics policy. We look forward to seeing your revised manuscript and we thank you for your interest in The Journal of Physiology!

REFeree COMMENTS

Referee #1:

This work suggests a novel role for neutrophil NET secretion in supporting GLUT4 translocation and insulin sensitivity following exercise which adds to a growing field of knowledge in which immune cells have distinct physiological roles in skeletal muscle health and homeostasis. While the premise is quite interesting, there are limitations to the experiments which dampen enthusiasm for the interpretations herein.

Major points

1. Fig 2C results: as written it's slightly unclear what's known and what the new data is (especially regarding sentence 1). More importantly, Cxcl1 and Il6 are listed as myokines, but they can also be produced by neutrophils, and as they decrease without neutrophils, they likely are being produced by the neutrophils. As I know Cxcl1 and Il6 can also be produced by muscle.. this can be discussed in the discussion if you see fit.
2. Fig4A: I'm struggling with the interpretation that low dispersed staining "likely represent(s) residual neutrophil-derived material". How can you be certain this is not background staining? At 1 hour post-EPS, this antibody in the bloodstream may label some blood neutrophils, but I anticipate neutrophil numbers are slightly down in the blood and circulating cells are less likely to get into the muscle at that time. Can you cite timecourse data to support your interpretation? Or provide additional information regarding how quickly neutrophils degranulate/ release NETs? It would be most helpful to inject QD655 anti-GR1 Ab at the same time as in Fig1&2 to ensure neutrophils were similarly recruited into the tissue. However, you at least need need to justify the decision to inject the antibody at 1 hr post.
3. Fig5: In order to interpret Sytox staining as NETs, is it known that this intensity of contraction does not cause damage to the muscle cells? Additionally, what is the justification for the use of these specific timepoints? It would seem that if low dose insulin+ contraction elicits the most GLUT4 translocation, you should then see more neutrophil staining and more NET staining between basal contraction and low-insulin contraction groups, but Fig4&5 data don't show that in the quantification... A timecourse with GLUT4 translocation, neutrophil, and NET staining or the use of a NET-release inhibitor would help support the interpretations herein.
4. Fig 6: if interested in neutrophil effects, shouldn't the +/- DNase conditions be the ones compared?
5. Fig 7: it is written that these images were taken in pre-contracted muscle fibers which is concerning as insulin alone doesn't seem to recruit neutrophils and thus, there should not be NET staining in these images. Is that accurate? And was this quantification only done for one image?
6. Fig 8: How can you be sure this DNA labeling is neutrophil derived and not muscle derived after 2 hours of forced running which does likely induce some muscle damage? Experiments blocking the release of NETs from neutrophils or somehow showing that the DNA is neutrophil-derived would enhance interpretations of this work.

Minor points

1. The following abbreviations require full spelling at initial mention: GAP, EPS, EDL
2. This is more my curiosity than required in the manuscript, but, why were mice fasted so long prior to muscle stim experiments?
3. Within the methodology, euthanasia can be assumed, but it would be helpful to state as methodology discussing live vs. postmortem experiments are listed sequentially. For example, under the wheel running model "before running session. After [euthanasia] and perfusion fixation, skeletal muscle..."
4. What dilution/concentration immunoblotting antibodies used at?
5. Fig1A: control spelled incorrectly
6. Fig1B: very cool imaging and the arrows are helpful for identification of neutrophils- is there any way to turn up the red on

the images for better visualization?

7. Fig3A: I was initially confused about this schematic because it showed much more than the data reported in B. As I see the schematic sets up the next few figures, I would move Fig3B to supplemental figures as it's previously published so that your neutrophil data is next to the schematic. These are different timepoints and you do write that, but seeing it visually is what's most helpful for many... I am still unsure where the 2DG data is?

8. Fig3B: "blood vessel" label should also say Evans Blue

9. Fig 4B. The low-insulin, post-contraction, overlay also does not look representative of the below quantification

10. Regarding "Notably, whereas QD signals were strong at 1 h after EPS (Figure 2), they had substantially declined by the total 2.5 h time point (2 h of resting plus 25~30 min after insulin stimulation), as reflected in the vertical axis scale (Figure 4B, left)"... I'm also sure you can compare AU from two different experiments on two different days at two different timepoints..

11. Fig 8: Can you include a schematic as timing has differed throughout the paper?

Referee #2:

Chen and Kanzaki investigated the role of neutrophils in post muscle contraction-mediated enhancement in insulin sensitivity. The study was prompted by prior work showing that neutrophils are implicated in glucose uptake evoked by muscle contraction. It has been known that neutrophils and other immune cells are critical for muscle recovery from injury. It is less well understood how local immune cells contribute to basic regulatory functions like glucose uptake surrounding feeding or exercise. The study is interesting. The authors used a unilateral contraction model by stimulating the sciatic nerve in one limb compared with sham non-contracting control limb. They then delivered a bolus of insulin or various agents to inhibit neutrophil recruitment or break apart NETs to determine GLUT4 membrane recruitment and glucose uptake. The main conclusion is that areas rich in neutrophil NETs have increased GLUT4 membrane localization after exercise and insulin stimulation. The interpretation is that this is one of the mechanisms that primes the muscle for enhanced glucose uptake in response to insulin after exercise. Insulin-induced phosphorylation of proteins involved in AKT signaling and GLUT movement were increased by prior contraction but were largely unaffected by NETs. This suggests that the mechanism may be an extracellular phenomenon. NETs, at least in the vasculature, are typically associated with decreased blood flow velocity.

Can the authors determine whether the NETs are within vessels or in the interstitial space? Does prior muscle contraction enhance extravasation of neutrophils into the interstitium, where neutrophils then generate NETs?

One looming question is whether the NETs, if in the muscle interstitium, interact with and remodel components of the extracellular matrix (ECM). Components of the ECM (collagens, fibronectin, laminin, etc.) are ligands for integrin receptors expressed directly on muscle cells that influence muscle glucose uptake through cytoskeleton rearrangement. Some discussion into the extramyocellular effects of these NETs needs to be added, because the evidence does not directly support that NETs promote an intrinsic mechanism in the muscle. At present the data support that NETs are associated with sarcolemmal GLUT4 localization. The mechanism explaining this connection is not clear. A NET-ECM-Integrin-muscle axis could be a plausible mechanism.

Figures

In each of the figures that contains microscopy images, it would be helpful to add a legend identifying the proteins that are being visualized (ie, GLUT4 (green), etc.).

Minor comment on approach

Does anesthesia influence the neutrophil recruitment dynamics surrounding muscle contraction, recovery, and insulin action?

END OF COMMENTS

Dear Editor,

We are pleased to resubmit our revised manuscript (**JP-RP-2025-290203R1**), **“Intramuscular neutrophil-derived immunometabolic niches locally boost insulin-responsive GLUT4 translocation after muscle contraction”** for consideration in *Journal of Physiology*. We thank the Reviewing Editor, Senior Editor, and reviewers for their constructive and insightful comments.

We have revised the manuscript accordingly and have addressed all comments in detail in the accompanying point-by-point response document. The revisions clarify the conceptual framework, explicitly acknowledge experimental boundaries and limitations, strengthen the integration with established mechanisms of microvascular insulin delivery and GLUT4 trafficking, and improve transparency in spatial quantification, figure presentation, and statistical reporting in full compliance with the journal’s policy.

We believe these revisions fully address all comments and strengthen the manuscript. We appreciate your time and consideration and hope the revised work will be suitable for publication in *Journal of Physiology*.

Sincerely,

Makoto Kanzaki, Ph.D.

Tohoku University

EDITOR COMMENTS

Reviewing Editor:

Comments to the Author:

This manuscript advances a clear conceptual shift: post-exercise insulin sensitization in skeletal muscle is proposed to be spatially gated by neutrophil-derived, NET-like extracellular DNA scaffolds laid down along the capillary meshwork, creating perivascular "immunometabolic niches" where physiological insulin preferentially drives GLUT4 translocation and glucose uptake. The claim is supported by paired in situ contraction and running models, two-photon/confocal imaging that localize GLUT4 enrichment to capillary-adjacent cfDNA structures, and DNase experiments that blunt low-insulin responses and attenuate AS160 (T642) phosphorylation, while sparing high-insulin effects-together positioning NET-rich perivascular topology as a structural determinant of insulin action after exercise.

Your DNase data convincingly show the necessity of extracellular DNA structures for low-dose insulin potentiation (GLUT4, 2-DG, AS160 T642), but do not establish sufficiency or exclude vascular delivery contributions. Could you please expand this point further?

RESPONSE

We appreciate the editor's insightful comment. We agree that our DNase experiments demonstrate the necessity of extracellular DNA/NET-like structures for potentiation of insulin action under low, physiological insulin concentrations, but do not establish their sufficiency nor exclude a contribution from vascular insulin delivery. We have revised the **Discussion** to explicitly acknowledge this limitation.

Specifically, we now state that disruption of extracellular DNA structures by DNase supports the necessity, but not the sufficiency, of NET-like structures for enhanced insulin action at low insulin levels. We further clarify that the present data cannot distinguish whether NETs directly potentiate insulin signaling at the myofiber surface or indirectly enhance insulin action by modulating local capillary insulin availability.

Importantly, we also emphasize that several observations are more consistent with a model of local insulin potentiation rather than a global enhancement of insulin delivery, including the selective DNase sensitivity observed at low but not high insulin doses, and

the spatial confinement of GLUT4 enrichment to NET-rich capillary regions. These additions are now explicitly discussed in the revised **Discussion**.

- It seems it is necessary to have an integrative paragraph mapping the NET-niche model onto two established axes: (i) microvascular insulin delivery/perfusion and (ii) GLUT4 trafficking control (AS160/TBC1D1).

RESPONSE

We thank the Editor for the suggestion to improve conceptual clarity. In response, we have revised the **Discussion** to streamline the description of the NET-based immunometabolic niche.

Specifically, we now more concisely describe the NET-rich perivascular niche as a spatially restricted site where vascular and myofiber-intrinsic mechanisms converge to promote localized GLUT4 enrichment in adjacent myofibers. We believe this revision improves clarity while preserving the integrative framework proposed in the study.

- All primary data are from male mice, fast-twitch EDL, anesthetized EPS, with a helpful validation in a running model. Please consider explaining these boundaries and broader claims (sex, species, slow-twitch SOL) as hypotheses.

RESPONSE

We thank the Editor for the insightful comment regarding the experimental boundaries and the scope of our conclusions. In response, we have revised the **Discussion** to explicitly state that all primary data in the present study were obtained from male mice using electrically evoked contractions under anesthesia in the predominantly fast-twitch EDL muscle. We now clearly acknowledge that the conclusions are bounded by this specific experimental context.

Furthermore, we have revised the text to explicitly frame potential extensions of our findings—across sex, species, and muscle fiber types (e.g., slow-twitch SOL muscle)—as hypotheses rather than established conclusions. We now emphasize that whether a similar neutrophil/NET-dependent mechanism operates in slow-twitch

muscle or exhibits sex- or species-specific differences remains unknown and requires direct experimental testing.

Finally, we clarify that the running experiments are presented as supportive evidence for the physiological relevance of NET formation under exercise conditions within the boundaries of the present study, while noting that broader generalizability across muscle types and exercise modalities remains to be determined.

We believe these revisions appropriately address the Editor's concerns and improve the clarity and rigor of the **Discussion**.

- Your signature strength is spatial quantification (ZY compression → line-profile AUC). This editor suggests promoting this workflow from legend-level detail to the main text and stating clearly why this transformation is appropriate for capillary-aligned signals. Provide the analysis script, probably as supplemental, as this will materially increase reader confidence and citations.

RESPONSE

We thank the Editor for highlighting the importance of our spatial quantification approach. In response, we have revised the manuscript to promote the axis-aligned ZY-compression-based workflow from legend-level detail to the main text and to clearly explain why this transformation is appropriate for capillary-aligned signals.

Specifically, we expanded the **Methods** section to provide a step-by-step, script-equivalent description of the image analysis procedure, including axis alignment, X-axis summation to generate Z–Y projections, subsequent Y-axis compression to obtain one-dimensional intensity profiles, and AUC-based quantification. This description explicitly explains how the approach preserves spatial continuity along the capillary axis while minimizing signal dilution from surrounding regions. Importantly, this workflow was implemented entirely using standard, built-in ImageJ/Fiji operations and did not rely on any custom analysis script, making it fully reproducible from the procedural description provided.

We also revised the **Results** section to explicitly reference this workflow and to emphasize how it enables detection of regionally restricted GLUT4 enrichment adjacent

to NET-rich regions under low-insulin conditions. In addition, we revised **Figure 6** to visually integrate representative XY images, Z–Y projections, one-dimensional compressed intensity images, and a schematic overview of the spatial quantification workflow. The updated figure legend now clearly describes each transformation step and its role in the analysis.

Together, these revisions clarify both the rationale and implementation of the spatial quantification strategy, enhance reproducibility without embedding code in the main text, and strengthen the presentation of this workflow as a core methodological and conceptual contribution of the study.

- I suggest adding a central schematic that aligns time and space: contraction → neutrophil recruitment → NET deposition along capillaries → low insulin → localized GLUT4 translocation; include the differential effects at AS160 (T642) vs. Akt (S473). This will let general readers "see" the mechanism in 10 seconds.

RESPONSE

In response to the reviewer's suggestion, we have added a central schematic that integrates both temporal and spatial dimensions of the proposed mechanism as Figure 8. This schematic visually aligns the sequence from muscle contraction to neutrophil recruitment, intravascular NET deposition along capillaries, and the resulting NET-dependent, localized enhancement of insulin-stimulated GLUT4 translocation. The figure also explicitly illustrates the differential signaling effects observed at AS160 (T642) versus Akt (S473).

Please consider clearly responding to this question in the discussion section: Why does a spatial, immune-governed mechanism matter for how one bout of exercise improves insulin efficiency at physiological insulin? Which could link to the translational relevance.

RESPONSE

We have revised the **Discussion** to directly address why a spatially confined, immune-governed mechanism is biologically and translationally relevant. Specifically, we now emphasize that post-exercise insulin sensitization can arise from localized

perivascular microdomains rather than uniform changes across muscle tissue. We also highlight that spatially restricted, neutrophil-derived IL-1 signaling within NET-rich niches may permit beneficial modulation of muscle insulin responsiveness while avoiding excessive inflammation. Together, this spatial organization provides a mechanistic explanation for how a single bout of exercise can substantially improve insulin efficiency at physiological insulin concentrations without requiring systemic hyperinsulinemia. We further discuss how impaired neutrophil recruitment or NET formation in insulin-resistant states could disrupt the establishment of these niches, potentially contributing to blunted exercise-induced insulin sensitization and underscoring translational relevance.

Senior Editor:

Comments for Authors to ensure the paper complies with the Statistics Policy:

If you choose to submit a revised manuscript for continued consideration, please visit the statistics policy to ensure compliance. The two main points that require revision are

1. reporting variability as SD rather than SEM and 2. reporting precise p-values.

Please do note that it is acceptable for p values less than 0.001, to simply report $p < 0.001$. All others need precise p-value reporting. Thank you.

RESPONSE

In accordance with the journal's Statistics Policy, we now report variability as mean \pm SD and provide exact adjusted p values for all post hoc comparisons indicated in the figures, which are presented in **Supplementary Table 1**.

REFeree COMMENTS

Referee #1:

This work suggests a novel role for neutrophil NET secretion in supporting GLUT4 translocation and insulin sensitivity following exercise which adds to a growing field of knowledge in which immune cells have distinct physiological roles in skeletal muscle health and homeostasis. While the premise is quite interesting, there are limitations to the experiments which dampen enthusiasm for the interpretations herein.

Major points

1. Fig 2C results: as written it's slightly unclear what's known and what the new data is (especially regarding sentence 1). More importantly, Cxcl1 and Il6 are listed as myokines, but they can also be produced by neutrophils, and as they decrease without neutrophils, they likely are being produced by the neutrophils. As I know Cxcl1 and Il6 can also be produced by muscle.. this can be discussed in the discussion if you see fit.

RESPONSE

We thank the reviewer for this constructive comment. We agree that CXCL1 and IL-6 can be produced not only by skeletal muscle fibers but also by recruited neutrophils. In our previous in vitro contraction model, which excludes immune cells, we confirmed that contractile activity robustly induces CXCL1 and IL-6 expression in muscle cells (Nedachi *et al.*, 2008; Nedachi *et al.*, 2009; Farmawati *et al.*, 2013). Thus, muscle fibers are clearly capable of producing these myokines upon contractile activity. However, in vivo, where multiple cell types coexist, including neutrophils, it is not possible to fully exclude contributions from non-muscle sources.

To clarify what is already known versus what is newly demonstrated in our study, and to address the reviewer's point, we have revised the corresponding portion of the **Results** section, accordingly.

2. Fig4A: I'm struggling with the interpretation that low dispersed staining "likely represent(s) residual neutrophil-derived material". How can you be certain this is not background staining? At 1 hour post-EPS, this antibody in the bloodstream may label some blood neutrophils, but I anticipate neutrophil numbers are slightly down in the

blood and circulating cells are less likely to get into the muscle at that time. Can you cite timecourse data to support your interpretation? Or provide additional information regarding how quickly neutrophils degranulate/ release NETs? It would be most helpful to inject QD655 anti-GR1 Ab at the same time as in Fig1&2 to ensure neutrophils were similarly recruited into the tissue. However, you at least need need to justify the decision to inject the antibody at 1 hr post.

RESPONSE

We appreciate the reviewer's insightful comments regarding the interpretation of the low, dispersed QD-GR1 signal in Fig. 4A (**new Fig. 3B**) and the rationale for administering the QD-GR1 antibody 1 h after EPS.

In Fig. 1, QD-GR1 antibody was administered immediately after EPS-evoked contractions, followed by a 1-h in vivo labeling period. After perfusion-fixation, circulating intravascular neutrophils were removed, ensuring that the QD-GR1 signal reflected neutrophils adherent to the muscle microvascular endothelium. These data demonstrate that neutrophils are robustly recruited to exercised muscle and remain adherent within the microvasculature during the first post-EPS hour.

In contrast, the experiment shown in new Fig. 3 was designed to evaluate post-exercise enhancement of insulin responsiveness. To maintain a labeling duration comparable to Fig. 1 while avoiding prolonged systemic exposure to GR1 antibody, QD-GR1 Ab was administered 1 h after EPS, followed by an additional 1-h in vivo labeling period prior to insulin stimulation. This timing was chosen because extended exposure to GR1 antibodies is known to induce neutropenia. After the labeling period, a low dose of insulin was administered for ~30 min, resulting in a total antibody exposure time of approximately ~1.5 h, a duration that minimizes GR1-induced neutropenia while still permitting effective labeling.

Importantly, based on the findings in Fig. 1, neutrophils have already been recruited to the exercised muscle and remain adherent within the microvasculature at 1 h post-EPS. Thus, administering QD-GR1 Ab at this time point allows labeling of these previously recruited neutrophils and any associated GR1-positive neutrophil-derived material. Following perfusion-fixation, circulating neutrophils were largely cleared, reducing the likelihood that the observed signal represents background staining.

To address this point, we have added clarifying statements to the **Methods** describing the rationale for administering QD-GR1 antibody 1 h after EPS and noting that this timing is expected to label neutrophils previously recruited to exercised muscle while minimizing prolonged systemic GR1 antibody exposure.

Furthermore, pretreatment with AZD+SB markedly suppressed EPS-induced neutrophil recruitment in Fig. 1. Consistent with this, the QD-GR1 signal in new Fig. 3B is substantially reduced under the same pretreatment conditions. This concordance across experiments strongly supports the interpretation that the faint, dispersed QD-GR1 signal reflects residual recruited neutrophils and/or neutrophil-derived material, rather than nonspecific background staining.

We also observed prominent Sytox Orange–positive cfDNA structures specifically in the exercised muscle (**new Fig. 4**). Importantly, these cfDNA signals were completely absent in mice rendered neutropenic by anti-GR1 antibody treatment initiated the day before the experiment, using the same depletion protocol as in Fig. 1. When the identical staining procedure used for new Fig. 3 was applied under these neutropenic conditions, no Sytox Orange–positive structures were detected. These new data are now included as **Supplementary Fig. S3** in the revised manuscript.

The complete loss of cfDNA signal in neutropenic mice strongly supports the conclusion that the Sytox Orange–positive structures originate from neutrophils rather than from nonspecific background staining. Together with these findings, the faint and dispersed QD-GR1 signal observed in new Fig. 3B is most parsimoniously interpreted as residual neutrophil-derived material remaining within the exercised muscle microvasculature.

Regarding the reviewer’s suggestion to inject QD-GR1 Ab at the same time point as in Fig. 1 and Fig. 2, we agree that such an approach would, in principle, provide a more direct comparison of neutrophil labeling. However, this experimental design is constrained by several factors: the need to maintain comparable in vivo antibody exposure durations across experiments, the risk of GR1 antibody–induced neutropenia during the prolonged post-EPS resting period required for insulin responsiveness measurements, and the difficulty in discriminating labeling of circulating neutrophils from those recruited to the tissue under these conditions. For these reasons, we believe that performing this experiment is not feasible within the scope of the present study.

Nonetheless, the consistent reduction of QD-GR1 Ab signals in the AZD+SB-treated group across experiments, together with the neutrophil dependence of cfDNA structures in exercised muscle, provides convergent evidence supporting our conclusion that the faint QD-GR1 signal in new Fig. 3B reflects neutrophil-derived material rather than background staining.

3. Fig5: In order to interpret Sytox staining as NETs, is it known that this intensity of contraction does not cause damage to the muscle cells? Additionally, what is the justification for the use of these specific timepoints? It would seem that if low dose insulin+ contraction elicits the most GLUT4 translocation, you should then see more neutrophil staining and more NET staining between basal contraction and low-insulin contraction groups, but Fig4&5 data don't show that in the quantification... A timecourse with GLUT4 translocation, neutrophil, and NET staining or the use of a NET-release inhibitor would help support the interpretations herein.

RESPONSE

We thank the reviewer for raising the important concern regarding whether the Sytox Orange-positive signals could arise from muscle damage rather than neutrophil-derived extracellular DNA.

Importantly, muscle injury is highly unlikely under the present EPS conditions. As a primary and highly sensitive assessment of vascular and sarcolemmal integrity, we evaluated Evans Blue Dye (EBD) extravasation, which is widely regarded as one of the most reliable in vivo methods for detecting muscle fiber membrane disruption and vascular leakage. As shown in **Supplementary Figures S1 and S2 (previously Figure 3A)**, EBD was strictly confined to the vasculature, with no evidence of dye extravasation into the muscle interstitium or myofibers. This indicates that the EPS protocol did not compromise vascular integrity or induce myofiber membrane damage. Accordingly, to explicitly state that no muscle injury occurred under the present EPS conditions, we have added a clear description of these results to the **Results** section of the revised manuscript.

The absence of EBD extravasation is particularly important in this context because Sytox Orange is membrane-impermeant and only labels extracellular DNA or DNA within cells with disrupted membranes. Therefore, if Sytox-positive signals were derived from injured myofibers, concomitant EBD uptake would be expected. The complete lack of EBD leakage thus provides strong evidence that the Sytox Orange-positive structures do not originate from damaged muscle fibers.

Most critically, the Sytox Orange-positive signals were completely abolished by intravascular administration of DNase I, which selectively degrades extracellular DNA within the circulation. This DNase I sensitivity demonstrates that the Sytox signal reflects extracellular DNA within the vasculature rather than DNA released by injured myofibers. Furthermore, these Sytox-positive structures were entirely absent in neutropenic mice generated by anti-GR1 treatment using the same depletion protocol as in Figure 1 (**Supplementary Figure S3**). Together, the DNase I sensitivity and neutrophil dependence provide compelling evidence that the observed Sytox Orange-positive structures originate from neutrophils and represent NET-associated extracellular DNA.

Regarding the choice of timepoints, our experimental design was guided by prior work demonstrating that contraction-induced intracellular signaling and GLUT4 translocation return to baseline within approximately 1 hour after stimulation (Tsuchiya et al., *Cell Reports*, 2018). Although that study used intramuscular electrical stimulation, it provides a well-established temporal reference for acute contraction-evoked signaling dynamics. For the reviewer's reference, we have included here the **relevant Supplementary Figure from that study (For Reviewers' Reference)**, which illustrates the temporal profile of EPS-evoked AMPK/TBC1D1/AS160 phosphorylation and GLUT4 translocation returning to pre-stimulation levels within 60 minutes. Accordingly, the EPS intensity and timing in the present study were selected to induce physiological muscle contraction while allowing recovery of these transient signaling events prior to assessing insulin responsiveness.

Finally, although we agree that additional time-course analyses or the use of pharmacological inhibitors of NET release would further enhance the mechanistic depth of the study, we believe that the convergence of evidence presented here—namely, the preserved vascular and sarcolemmal integrity confirmed by EBD extravasation, the DNase I sensitivity of the extracellular structures, and their complete dependence on the presence of neutrophils—provides strong support for interpreting the Sytox Orange-

positive signals as neutrophil-derived extracellular DNA rather than as artifacts arising from muscle damage or nonspecific nuclear staining.

For Reviewers' Reference

Figure S4

FigureS4. In situ muscle contraction model applicable to imaging analysis of GLUT4 translocation (Tsuchiya et al., *Cell Reports*, 2018).

(A) Scheme for in situ muscle contraction model using EPS. Note that imaging region is apart from the positions of injecting electrodes. (B) Protocol of EPS application. Arrows

represent the time for acquiring the images or samples for Western blotting. (C) Snapshot of myc-GLUT4-EGFP fluorescence in QFM observed with two-photon excitation microscopy. (D) Intensity profile of myc-GLUT4-EGFP fluorescence across the Sarcolemma (whitebox in(C)). Intensities from which the mean intensity in the cytosol was subtracted were shown(ΔF). We defined sarcolemma GLUT4 as the peak ΔF . (E) Power spectrum of the image shown in (C) obtained by 2D-FFT. By using these spectra, we calculated mean radial profiles of the power around concentric circles as a function of spatial frequencies. (F) Mean radial profile of the power between spatial Frequencies of $0.4\text{--}0.6\mu\text{m}^{-1}$ of the image shown in (C). Baseline-subtracted power (ΔP) is shown. We defined T-tubule GLUT4 as the peak amplitude of ΔP within this range. (G) Changes in sarcolemma (upper) and T tubule (lower) GLUT4 in response to EPS-evoked contraction in QFMs. Data are normalized to the mean value of before EPS, and lines and error bars represent mean and SEM, respectively. Statistical analyses were performed with Dunnett's multiple comparison. * $P < 0.05$. (H) Western blotting analyses after EPS-evoked contraction in QFMs. (Tsuchiya et al., *Cell Reports*, 2018)

4. Fig 6: if interested in neutrophil effects, shouldn't the +/- DNase conditions be the ones compared?

RESPONSE

We agree with the reviewer that comparisons between the -DNase and +DNase conditions are the most relevant for evaluating neutrophil-dependent effects. In the original version of Fig. 6, these comparisons were just indicated with “#” symbols, but we acknowledge that this presentation was not sufficiently clear.

To improve clarity, we have revised some of Fig. 6B (**new Figure 5B**) so that the statistical comparisons between the -DNase and +DNase groups are now explicitly connected with brackets/lines in the updated graphs (e.g., AS160T642, AktS473). This makes the DNase-dependent differences immediately visible and easier for the reader to interpret.

In accordance with the journal's Statistics Policy, we now report variability as mean \pm SD and provide exact adjusted p values for all post hoc comparisons indicated in the figures, which are presented in **Supplementary Table 1**.

5. Fig 7: it is written that these images were taken in pre-contracted muscle fibers which is concerning as insulin alone doesn't seem to recruit neutrophils and thus, there should not be NET staining in these images. Is that accurate? And was this quantification only done for one image?

RESPONSE

We appreciate the reviewer's careful evaluation of our figure. To address the concern, we would like to clarify that the images shown in Fig. 7 (**now Figure 6**) were obtained from post-contraction muscles, as explicitly stated in the figure legend: "***NET-associated regional enhancement of low-insulin-induced GLUT4 translocation in post-contraction muscles.***" Thus, these images were not acquired from pre-contracted or resting muscle. Because the EPS-induced contraction had already occurred before insulin application, neutrophil recruitment and NET formation were already present at the time of imaging. Accordingly, the detection of Sytox-positive NET structures in these images is an expected outcome of the prior contraction stimulus, and insulin alone does not induce neutrophil recruitment or NET release in our model.

To further avoid any potential misunderstanding, we have also revised the **Figure 6 legend** to explicitly restate that the images were acquired after completion of the EPS-induced contraction protocol. This clarification ensures that readers immediately recognize that the observed NET formation resulted from the prior contraction stimulus rather than from insulin treatment.

Regarding quantification, Fig. 7 (**new Figure 6B**) illustrates the pixel-wise GLUT4 and Sytox intensity profiles derived from the representative image shown in new Figure 6A. In contrast, new **Figure 6C** presents the area-under-the-curve (AUC) analysis compiled from three independent experiments, comprising eight total regions of interest. For each region, both the Sytox-positive (NET-rich) portion and the Sytox-negative (NET-poor) portion were quantified, and the paired AUC values were compared within each region. Thus, the quantification is not based on a single image, but on pooled, region-matched measurements across multiple biological replicates.

6. Fig 8: How can you be sure this DNA labeling is neutrophil derived and not muscle derived after 2 hours of forced running which does likely induce some muscle damage? Experiments blocking the release of NETs from neutrophils or somehow showing that the DNA is neutrophil-derived would enhance interpretations of this work.

RESPONSE

We appreciate the reviewer's important question regarding the origin of the extracellular DNA observed after the 2-hour treadmill running protocol. Importantly, our running protocol is identical to the previously established condition reported by Tsuchiya et al. (*Cell Reports*, 2018), in which hematoxylin and eosin staining demonstrated normal skeletal muscle histology in both sedentary and 2-hour-exercised muscles, with no evidence of myofiber damage. Thus, this paradigm is widely regarded as a physiological, non-injurious contraction model.

Moreover, the Sytox Orange-positive signals were completely abolished by intravascular administration of DNase I, which selectively degrades extracellular DNA within the circulation. This DNase I sensitivity demonstrates that the Sytox signal reflects extracellular DNA within the vasculature rather than DNA released by injured myofibers. Therefore, the Sytox-positive extracellular DNA signal is highly unlikely to originate from disrupted myofibers and instead is most consistent with neutrophil-derived NETs.

We agree that future studies employing pharmacological inhibitors or genetic suppression of NET release would further strengthen the mechanistic interpretation, and we view this as an important direction for continued investigation.

Minor points

1. The following abbreviations require full spelling at initial mention: GAP, EPS, EDL

RESPONSE

We appreciate the reviewer's helpful suggestion regarding abbreviation usage. In the revised manuscript, we have spelled out all relevant terms at their first appearance, including GTPase-activating protein (GAP), electrical pulse stimulation (EPS),

and extensor digitorum longus (EDL) muscle, followed by their abbreviations in parentheses. Abbreviations are consistently used thereafter throughout the text.

2. This is more my curiosity than required in the manuscript, but, why were mice fasted so long prior to muscle stim experiments?

RESPONSE

Thank you for your thoughtful question. In our recent study investigating contraction-dependent improvements in insulin responsiveness using CEFIP-deficient mice (Nyasha *et al.*, 2025), we employed an overnight fasting protocol to standardize the metabolic baseline prior to muscle stimulation. To ensure consistency and allow direct comparison with those findings, we adopted the same fasting duration in the present study. Accordingly, an overnight fasting period was used to maintain comparable metabolic conditions across studies.

3. Within the methodology, euthanasia can be assumed, but it would be helpful to state as methodology discussing live vs. postmortem experiments are listed sequentially. For example, under the wheel running model "before running session. After [euthanasia] and perfusion fixation, skeletal muscle..."

RESPONSE

Thank you for this helpful comment. We agree that it is important to clearly distinguish in vivo procedures from postmortem processing in the Methods section. Accordingly, we have revised the Methods to explicitly state that, after completion of the running protocol, mice were euthanized and immediately subjected to perfusion fixation, and that in some experiments Evans blue dye was administered intravenously under anesthesia immediately prior to perfusion fixation. This clarification has now been added to the **Methods** section of the revised manuscript.

4. What dilution/concentration immunoblotting antibodies used at?

RESPONSE

Thank you for pointing out the need to specify the antibody dilutions. We have now added detailed information to the Methods section. Specifically, commercial antibodies were used at a 1:2000 dilution, while the laboratory-generated antibodies were used at a final concentration of 0.5 µg/mL. For detection, HRP-conjugated secondary antibodies were used at a 1:20,000 dilution, including Goat anti-Rabbit IgG (H+L) (Thermo Fisher, #32460) and Goat anti-Mouse IgG (H+L) (Thermo Fisher, #32430). These details are now clearly described in the revised manuscript.

5. Fig1A: control spelled incorrectly

RESPONSE

Thank you for pointing this out. This error in Figure 1 has been corrected in the revised Figure.

6. Fig1B: very cool imaging and the arrows are helpful for identification of neutrophils- is there any way to turn up the red on the images for better visualization?

RESPONSE

Thank you for your helpful suggestion. We have enhanced the red fluorescence signal (*upper panels*) uniformly across all relevant images to improve the visualization of neutrophils in the revised **Figure 1B**.

7. Fig3A: I was initially confused about this schematic because it showed much more than the data reported in B. As I see the schematic sets up the next few figures, I would move Fig3B to supplemental figures as it's previously published so that your neutrophil data is next to the schematic. These are different timepoints and you do write that, but seeing it visually is what's most helpful for many... I am still unsure where the 2DG data is?

RESPONSE

Thank you very much for this helpful suggestion and for pointing out the source of potential confusion. We agree that the flow of the figures becomes clearer when the neutrophil data directly follows the schematic. Accordingly, we have moved **Fig. 3B** to

the **Supplementary Figure S2**, and the revised **Fig. 3** now presents the neutrophil results immediately after the schematic.

We agree that this text may have contributed to the confusion. This section summarizes our previously published work describing the temporal profile of contraction-induced 2DG uptake (Nyasha *et al.*, 2025). In the present study, 2DG uptake data are generated only in the context of the DNase-treatment experiments (**new Figure 5**). We have revised the **Results** section to clearly distinguish previously published findings from the new experimental data presented in this manuscript.

We hope that these revisions improve the clarity and logical flow of the manuscript.

8. Fig3B: "blood vessel" label should also say Evans Blue

RESPONSE

Thank you for pointing this out. We have added “Evans Blue” to the blood vessel label and moved this panel to the **Supplementary Figure S2**, as described above.

9. Fig 4B. The low-insulin, post-contraction, overlay also does not look representative of the below quantification

RESPONSE

Thank you for this helpful comment. We agree that the overlay images made it difficult to appreciate the spatial localization of neutrophils relative to GLUT4. To improve clarity, we replaced the overlay with separate single-channel images and indicated the positions of representative neutrophils using arrowheads (**new Figure 3B**). This presentation more clearly reflects the quantified results and allows easier visualization of contraction-induced neutrophil recruitment.

10. Regarding "Notably, whereas QD signals were strong at 1 h after EPS (Figure 2), they had substantially declined by the total 2.5 h time point (2 h of resting plus 25~30 min after insulin stimulation), as reflected in the vertical axis scale (Figure 4B, left)"...

I'm also sure you can compare AU from two different experiments on two different days at two different timepoints..

RESPONSE

We agree with the reviewer's concern and have therefore removed this statement from the **Results** section to avoid any implication of quantitative or qualitative comparison between independent experiments performed at different time points.

11. Fig 8: Can you include a schematic as timing has differed throughout the paper?

RESPONSE

Thank you for this helpful suggestion. We agree that the timing of the different experiments may not be immediately intuitive when presented separately. To improve clarity, we have added a schematic timeline to **new Figure 7 (previously Figure 8)** summarizing the experimental sequence and the specific time points.

Referee #2:

Chen and Kanzaki investigated the role of neutrophils in post muscle contraction-mediated enhancement in insulin sensitivity. The study was prompted by prior work showing that neutrophils are implicated in glucose uptake evoked by muscle contraction. It has been known that neutrophils and other immune cells are critical for muscle recovery from injury. It is less well understood how local immune cells contribute to basic regulatory functions like glucose uptake surrounding feeding or exercise. The study is interesting. The authors used a unilateral contraction model by stimulating the sciatic nerve in one limb compared with sham non-contracting control limb. They then delivered a bolus of insulin or various agents to inhibit neutrophil recruitment or break apart NETs to determine GLUT4 membrane recruitment and glucose uptake. The main conclusion is that areas rich in neutrophil NETs have increased GLUT4 membrane localization after exercise and insulin stimulation. The interpretation is that this is one of the mechanisms that primes the muscle for enhanced glucose uptake in response to insulin after exercise. Insulin-induced phosphorylation of proteins involved in AKT signaling and GLUT movement were increased by prior contraction but were largely unaffected by NETs. This suggests that the mechanism may be an extracellular phenomenon. NETs, at least in the vasculature, are typically

associated with decreased blood flow velocity.

Can the authors determine whether the NETs are within vessels or in the interstitial space? Does prior muscle contraction enhance extravasation of neutrophils into the interstitium, where neutrophils then generate NETs?

RESPONSE

We administered DNase systemically via tail-vein injection, which almost completely abolished the contraction-dependent (**new Figure 4**) and exercise-dependent (**new Figure 7**) Sytox Orange–positive signals (cfDNA). Because intravenously delivered DNase primarily accesses the intravascular compartment, the marked loss of these signals strongly suggests that the cfDNA/NETs detected in our experiments are predominantly located within the vasculature.

The structural integrity of the microvascular network following contraction was confirmed by Evans Blue staining (**Supplementary Figures S1 and S2**), demonstrating that neither sciatic-nerve EPS or 2 h of running caused overt vascular damage or leakage. This further supports the interpretation that DNase effectively degraded intravascular NETs rather than cfDNA originating from interstitial tissue injury.

Consistent with this conclusion, Sytox Orange signals were absent when neutrophils were depleted (**Supplementary Figure S2**) or when mice were pretreated with AZD + SB, indicating that the detected cfDNA is neutrophil-derived and not a consequence of skeletal muscle damage.

Because Sytox Orange does not readily permeate intact vasculature, the signals observed under control conditions most likely reflect NET formation within the vascular lumen rather than NETs generated by neutrophils that have extravasated into the interstitial space or originating from damaged myofibers. Taken together, these findings argue against enhanced neutrophil extravasation and interstitial NET formation following muscle contraction in our experimental model.

One looming question is whether the NETs, if in the muscle interstitium, interact with and remodel components of the extracellular matrix (ECM). Components of the ECM (collagens, fibronectin, laminin, etc.) are ligands for integrin receptors expressed

directly on muscle cells that influence muscle glucose uptake through cytoskeleton rearrangement. Some discussion into the extramyocellular effects of these NETs needs to be added, because the evidence does not directly support that NETs promote an intrinsic mechanism in the muscle. At present the data support that NETs are associated with sarcolemmal GLUT4 localization. The mechanism explaining this connection is not clear. A NET-ECM-Integrin-muscle axis could be a plausible mechanism.

RESPONSE

We thank the reviewer for this insightful comment regarding the localization of NETs and their potential functional significance. While our data indicate that NETs induced by muscle contraction or exercise are predominantly localized within the vasculature, we agree that their regional localization raises important questions regarding their potential local biological effects.

In response to this comment, we have substantially expanded the **Discussion** to address possible mechanisms by which regionally localized intravascular NET formation could influence the perivascular and muscle microenvironment. Specifically, we now discuss that NETs carry neutrophil-derived proteases (e.g., NE, MPO, PR3) that retain catalytic activity, can modulate extracellular matrix components and endothelial structures, and are capable of processing and activating IL-1 family cytokines. In addition, we have added a brief discussion of integrin-mediated mechanisms, noting that NET-associated modulation of the perivascular extracellular matrix could influence integrin-dependent adhesion and signaling at the muscle–vascular interface.

We further integrate these concepts with our previous findings demonstrating an essential role for neutrophil-derived IL-1 in muscle function, GLUT4 translocation, and contraction-induced glucose uptake (Tsuchiya *et al.*, 2018), and propose that NET-associated enzymes may contribute to spatially restricted, context-appropriate signaling at the muscle–vascular interface without requiring neutrophil extravasation into the interstitial space.

We believe that this expanded discussion clarifies the potential physiological relevance of intravascular NETs while appropriately acknowledging the need for future mechanistic studies.

Figures

In each of the figures that contains microscopy images, it would be helpful to add a legend identifying the proteins that are being visualized (ie, GLUT4 (green), etc.).

RESPONSE

Thank you for this constructive suggestion. We have revised the figures to clearly indicate the proteins visualized in each microscopy panel by adding color-coded labels (e.g., GLUT4 in green, etc.). These additions improve clarity and allow readers to easily identify the targets being observed. Corresponding descriptions have also been incorporated into the figure legends.

Does anesthesia influence the neutrophil recruitment dynamics surrounding muscle contraction, recovery, and insulin action?

RESPONSE

We acknowledge the reviewer's point regarding the potential influence of anesthesia on immune and metabolic responses. In our experimental design, all mice were maintained under identical and continuous anesthetic conditions throughout the electrical stimulation protocol. Importantly, the contraction stimulus was applied unilaterally, allowing the contralateral limb to serve as an internal control within the same animal. This paired comparison effectively controls for any systemic effects of anesthetic environment. Therefore, the differences observed between limbs can be attributed to the contraction stimulus itself rather than to anesthetic. We have added a statement to the **Methods** section in the revised manuscript to clarify this point.

Reference

- Farmawati A, Kitajima Y, Nedachi T, Sato M, Kanzaki M & Nagatomi R. (2013). Characterization of contraction-induced IL-6 up-regulation using contractile C2C12 myotubes. *Endocr J* **60**, 137-147.
- Nedachi T, Fujita H & Kanzaki M. (2008). Contractile C2C12 myotube model for studying exercise-inducible responses in skeletal muscle. *American journal of physiology Endocrinology and metabolism* **295**, E1191-1204.
- Nedachi T, Hatakeyama H, Kono T, Sato M & Kanzaki M. (2009). Characterization of contraction-inducible CXC chemokines and their roles in C2C12 myocytes. *American journal of physiology Endocrinology and metabolism* **297**, E866-878.
- Nyasha MR, Tachikawa J, Komatsuzaki H, Chen W, Onodera M, Kojima D, Yaoita F & Kanzaki M. (2025). CEFIP deficiency in mice enhances glucose tolerance despite compromised muscle function. *American journal of physiology Endocrinology and metabolism* **328**, E1021-E1040.
- Tsuchiya M, Sekiai S, Hatakeyama H, Koide M, Chaweewannakorn C, Yaoita F, Tan-No K, Sasaki K, Watanabe M, Sugawara S, Endo Y, Itoi E, Hagiwara Y & Kanzaki M. (2018). Neutrophils Provide a Favorable IL-1-Mediated Immunometabolic Niche that Primes GLUT4 Translocation and Performance in Skeletal Muscles. *Cell Rep* **23**, 2354-2364.

Dear Dr Kanzaki,

Re: JP-RP-2025-290203R1 **"Intramuscular neutrophil-derived immunometabolic niches locally boost insulin-responsive GLUT4 translocation after muscle contraction"** by Weijian Chen and Makoto Kanzaki

Thank you for submitting your manuscript to The Journal of Physiology. It has been assessed by a Reviewing Editor and by 2 expert referees and we are pleased to tell you that it is acceptable for publication following satisfactory revision.

REVISION CHECKLIST:

We look forward to receiving your revised submission.

Yours sincerely,

Karyn Hamilton
Senior Editor
The Journal of Physiology

REQUIRED ITEMS

- Papers must comply with the Statistics Policy: https://jp.msubmit.net/cgi-bin/main.plex?form_type=display_requirements#statistics.

In summary:

- If n {less than or equal to} 30, all data points must be plotted in the figure in a way that reveals their range and distribution. A bar graph with data points overlaid, a box and whisker plot or a violin plot (preferably with data points included) are acceptable formats.
- If $n > 30$, then the entire raw dataset must be made available either as supporting information, or hosted on a not-for-profit repository, e.g. FigShare, with access details provided in the manuscript.
- 'n' clearly defined (e.g. x cells from y slices in z animals) in the Methods. Authors should be mindful of pseudoreplication.
- All relevant 'n' values must be clearly stated in the main text, figures and tables.
- The most appropriate summary statistic (e.g. mean or median and standard deviation) must be used. Standard Error of the Mean (SEM) alone is not permitted.
- Exact p values must be stated. Authors must not use 'greater than' or 'less than'. Exact p values must be stated to three significant figures even when 'no statistical significance' is claimed.

- Please include an Abstract Figure file and an Abstract Figure legend. An appropriate figure legend, which should not exceed 150 words in length, should be included in the main manuscript file. The Abstract Figure is a piece of artwork designed to give readers an immediate understanding of the research and should summarise the main conclusions. If possible, the image should be easily 'readable' from left to right or top to bottom. It should show the physiological relevance of the manuscript so readers can assess the importance and content of its findings. Abstract Figures should not merely recapitulate other figures in the manuscript. Please try to keep the diagram as simple as possible and without superfluous information that may distract from the main conclusion(s). Abstract Figures must be provided by authors no later than the revised manuscript stage and should be uploaded as a separate file during online submission labelled as File Type 'Abstract Figure'. Please also ensure that you include the figure legend in the main article file. All Abstract Figures should be created using BioRender. Authors should use The Journal's premium BioRender account to export high-resolution images. Details on how to use and access the premium account are included as part of this email.

EDITOR COMMENTS

Reviewing Editor:

Comments to the Author:

The manuscript still needs to be aligned with the Journal of Physiology formatting guidelines. Specifically, the authors must ensure all data are reported as SD rather than SEM, and that exact p-values are required.

Senior Editor:

Comments for Authors to ensure the paper complies with the Statistics Policy:

As mentioned in my comments on your first version of the manuscript, please visit the statistics policy to ensure compliance. The two main points that require revision are 1. reporting variability as SD rather than SEM (I see that the figure legends are revised to indicate SD, but the methods section still indicates SE) and 2. reporting precise p-values. Please do note that it is acceptable for p values less than 0.001, to simply report $p < 0.001$. All others need precise p-value reporting. I noticed that you refer to precise p-values included in a supplementary table. I didn't see the table, but I also do not think this is an acceptable way to report the precise p-values. I recommend you simply add them to the figures themselves. Thank you very much.

Comments to the Author:

We appreciate the careful revisions you made to your manuscript. Only a few points remain unaddressed and we would like to provisionally accept the manuscript, pending acceptable revision to address these remaining points. Please carefully address Referee #1's remaining points. Also, please re-visit The Journal's statistics policy to ensure full compliance. The two main points that require revision are 1. reporting variability as SD rather than SEM (I see that the figure legends are revised to indicate SD, but the methods section still indicates "SE") and 2. reporting precise p-values. Please do note that it is acceptable for p values less than 0.001, to simply report $p < 0.001$. All others need precise p-value reporting as per the statistics policy. I noticed that you refer to precise p-values included in a supplementary table. I didn't see the table, but I also do not think this is an acceptable way to report the precise p-values. I recommend you simply add them to the figures themselves. Finally, please note The Journal's policy on supplemental information. As there are no restrictions on page length, The Journal does not usually accept supplemental/supporting material that cannot be included with the body of the manuscript. Please read The Journal's policy regarding Supporting Information and contact The Journal staff with specific questions you might have regarding this policy. Thank you in advance and we look forward to seeing your revised manuscript!

REFeree COMMENTS

Referee #1:

Thank you to the authors for their hard work on these revisions. Most of the explanations and associated revisions confirm that the details of the experiments were well thought out and most of the data now support the conclusions. I only have two comments:

1. I apologize if my incomplete sentence in the previous review threw you off, however, the edits regarding gene expression for Cxcl1 and Il6 for Fig6C have not improved the manuscript. I still believe that because both contraction conditions had muscle contractions, yet only one had neutrophils and Cxcl1/ Il6, that Cxcl1/Il 6 are due to neutrophil recruitment and not inherent muscle contraction (unless you have data to suggest that AZD+SB reduced contractility of the muscle or has additional off target effects). I expect that due to your previous work you may still disagree. However, I don't think you need to include substantial new text or significant discussion within the text as this is not a primary finding of the paper. I would remove all revised text in this section and revise from the initial submission as follows:

"Similar to our previous reports in a voluntary running model (Tsuchiya 2018), contraction increased mRNA levels of the neutrophil-associated gene Ly6G, the myokines/cytokines CXCL1 and IL-6, and the neutrophil granule protein MPO. Importantly, all four genes were almost completely abolished by AZD + SB treatment (Figure 2C)."

5. While the figure legend specifies this data was post-contraction, the results specify pre-contraction ("To this end, we analyzed pre-contraction muscles exposed to low-dose insulin."). Please update.

Referee #2:

The authors have done a nice job addressing my concerns. I have no further queries.

END OF COMMENTS

Dear Reviewing Editor, Senior Editor, and Reviewers,

We sincerely thank the Editors and Reviewers for their careful re-evaluation of our manuscript and for the constructive and helpful comments. We are grateful for the provisional acceptance and have carefully revised the manuscript to fully address all remaining points and to ensure complete compliance with *The Journal of Physiology* formatting and statistics policies. Our responses are detailed below.

We have now fully aligned the manuscript with *The Journal of Physiology* statistics policy.

1. All data throughout the manuscript are now reported as mean \pm SD rather than SEM. Importantly, we have corrected the Methods section, which previously referred to SE, to consistently indicate SD, ensuring uniform reporting across the manuscript.
2. We now report exact p-values directly in the Figure 7, in accordance with the Journal's statistics policy. Consequently, the previously mentioned supplementary table containing p-values has been removed, as all statistical information is now included within the main figures.
3. In accordance with the Journal's policy on Supporting Information, Supplementary Figures S1–S3 have been incorporated into the main manuscript. Specifically, Supplementary Figures S1 and S2 are now included as new Figure 4, and Supplementary Figure S3 is now included as new Figure 6.

We believe these changes ensure full compliance with the Journal's guidelines.

Response to Referee #1

We thank Referee #1 for the thoughtful feedback and for acknowledging the overall improvement of the manuscript.

We appreciate the reviewer's clarification and understand the concern regarding the interpretation of *Cxcl1* and *Il6* expression in relation to muscle contraction versus neutrophil recruitment.

As suggested, we have removed all revised text added in the previous revision and reverted this section to the wording from the initial submission, with only minor stylistic edits for clarity.

We thank the reviewer for pointing out this inconsistency. This discrepancy was due to a wording error. The Results section has now been corrected to consistently indicate that the data were obtained post-contraction, in agreement with the figure legend.

Dear Professor Kanzaki,

Re: JP-RP-2026-290203R2 "**Intramuscular neutrophil-derived immunometabolic niches locally boost insulin-responsive GLUT4 translocation after muscle contraction**" by Weijian Chen and Makoto Kanzaki

We are pleased to tell you that your paper has been accepted for publication in The Journal of Physiology.

Yours sincerely,

Karyn Hamilton
Senior Editor
The Journal of Physiology

IMPORTANT POINTS TO NOTE FOLLOWING ACCEPTANCE OF YOUR PAPER:

- **IMPORTANT NOTICE ABOUT OPEN ACCESS:** To assist authors whose funding agencies mandate immediate public access to published research findings, The Journal of Physiology allows authors to pay an Open Access (OA) fee to have their papers made freely available immediately on publication.

- You can help your research get the attention it deserves! Check out Wiley's free Promotion Guide for best-practice recommendations for promoting your work at: www.wileyauthors.com/eeo/guide. You can learn more about Wiley Editing Services which offers professional video, design, and writing services to create shareable video abstracts, infographics, conference posters, lay summaries, and research news stories for your research at: www.wileyauthors.com/eeo/promotion.

- If you would like to receive our 'Research Roundup', a monthly newsletter highlighting the cutting-edge research published in The Physiological Society's family of journals (The Journal of Physiology, Experimental Physiology, Physiological Reports, The Journal of Nutritional Physiology and The Journal of Precision Medicine: Health and Disease), please click this link, fill in your name and email address and select 'Research Roundup': <https://www.physoc.org/journals-and-media/membernews>

EDITOR COMMENTS

Reviewing Editor:

Comments to the Author:

We thank the authors for their careful revisions and clear responses to the reviewers' comments.

Senior Editor:

Comments to the Author:

Thank you for submitting your revised manuscript. We are pleased to accept it for publication in The Journal of Physiology.
Thank you for your interest in The Journal and Congratulations!

REFEREE COMMENTS

Referee #1:

The revisions are satisfactory, thank you.